# Developmental self-reactivity determines pathogenic Tc17 differentiation potential of naive CD8+ T cells in murine models of inflammation

Gil-Woo Lee[1,2,3], Young Ju Kim[1,2,3,4], Sung-Woo Lee[1,2,3], Hee-Ok Kim[5], Daeun Kim[5], Jiyoung Kim[6], You-Me Kim [6], Keunsoo Kang [7], Joon Haeng Rhee [1,2,4], Ik Joo Chung[3,8], Woo Kyun Bae[2,8], In-Jae Oh[8], Deok Hwan Yang [8] & Jae-Ho Cho [1,2,3,4] ✉

The differentiation of naive CD8+ T cells into effector cells is important for establishing immunity. However, the effect of heterogeneous naive CD8+ T cell populations is not fully understood. Here, we demonstrate that steady-state naive CD8+ T cells are composed of functionally heterogeneous subpopulations that differ in their ability to differentiate into type 17 cytotoxic effector cells (Tc17) in a context of murine inflammatory disease models, such as inflammatory bowel disease and graft-versus-host disease. The differential ability of Tc17 differentiation is not related to T-cell receptor (TCR) diversity and antigen specificity but is inversely correlated with self-reactivity acquired during development. Mechanistically, this phenomenon is linked to differential levels of intrinsic TCR sensitivity and basal Suppressor of Mothers Against Decapentaplegic 3 (SMAD3) expression, generating a wide spectrum of Tc17 differentiation potential within naive CD8+ T cell populations. These findings suggest that developmental self-reactivity can determine the fate of naive CD8+ T cells to generate functionally distinct effector populations and achieve immense diversity and complexity in antigen-specific T-cell immune responses.

The cytotoxic effector function of naive CD8+ T cells requires antigen-specific activation, proliferation, and differentiation, through coordinated expression of genes, driven by various transcriptional and epigenetic regulators[1]. Single-cell RNA-sequencing (scRNA-seq) data reveal that these effector cells are highly heterogeneous[2–4] and that individual effector cells can have distinct differentiation fates at the single-cell level. In fact, functionally diverse effector cells can be formed by differentiation of one cell in a single naive CD8+ T cell adoptive transfer experiment[3,4]. These studies provide credence to the prevailing notion that complex interactions of various factors during antigen-specific expansion and differentiation lead to heterogeneity of effector cell populations.

[1]Department of Microbiology and Immunology, Chonnam National University Medical School, Hwasun, Korea. [2]Medical Research Center for Combinatorial Tumor Immunotherapy, Chonnam National University Medical School, Hwasun, Korea. [3]Immunotherapy Innovation Center, Chonnam National University Medical School, Hwasun, Korea. [4]BioMedical Sciences Graduate Program, Chonnam National University Medical School, Hwasun, Korea. [5]Selecxine, Seoul, Korea. [6]Graduate School of Medical Science and Engineering, Korea Advanced Institute of Science and Technology, Daejeon, Korea. [7]Department of Microbiology, College of Science & Technology, Dankook University, Cheonan, Korea. [8]Department of Internal Medicine, Hwasun Hospital, Chonnam National University Medical School, Hwasun, Korea. ✉e-mail: jh_cho@jnu.ac.kr

Although the heterogeneity of effector cells has been relatively well studied, that of naive CD8$^+$ T cell population has not received much attention. However, recent studies have shown that naive CD8$^+$ T cells can be distinguished into subpopulations with distinct phenotypic and functional differences[5–10]. These differences can be identified based on relative differences in the intrinsic self-reactivity, that is, in the levels of CD5 expression, of naive CD8$^+$ T cells. For example, in an experimental setting of acute viral infections, cells with a high expression of CD5 (CD5$^{hi}$) exhibited a significantly increased antigen-specific responses compared with cells exhibiting a low expression of CD5 (CD5$^{lo}$)[5,6]. Moreover, in a similar virus infection model, a difference in effector differentiation fate was observed, with skewing CD5$^{hi}$ cells toward short-lived effector cells and CD5$^{lo}$ cells toward memory-precursor effector cells[5]. Although these findings are indicative of functional heterogeneity within naive CD8$^+$ T cell pools, the exact nature of cell-intrinsic properties and their mechanistic relationship with differential self-reactivity remain to be addressed and validated, not only in viral infection models but also in other immunological contexts.

Here, we investigated possible heterogeneity of naive CD8$^+$ T cells, particularly in a context of inflammatory disease models, such as inflammatory bowel disease (IBD) and graft-versus-host disease (GVHD), where CD8$^+$ T cells have been reported to differentiate into type 17 cytotoxic (Tc17) effector cells and cause immunopathological symptoms in mice and humans[11–16]. We demonstrate a crucial role of developmental self-reactivity of naive CD8$^+$ T cells in determining pathogenic Tc17 differentiation fates and provide an insight into the mechanism of how heterogeneous T cell immunity can be shaped in a steady-state condition even before antigen encounter.

## Results

### CD8$^+$ T$_N$ cells are composed of distinct subsets that differ in their capacity to induce immunopathology under inflammatory conditions

Peripheral naive CD8$^+$ T (T$_N$) cells were shown to be categorized into three distinct subsets, namely CD5$^{lo}$Ly6C$^-$, CD5$^{hi}$Ly6C$^-$, and CD5$^{hi}$Ly6C$^+$

cells, mainly based on differences in the intrinsic self-reactivity, which exhibited distinctly different functional behaviors upon viral infection[5]. To further explore the breadth of the physiological significance of individual CD8$^+$ T$_N$ subsets, we investigated the ability of these three distinct T$_N$ subpopulations to induce pathological inflammatory responses. To this end, CD44$^{lo}$ CD8$^+$ T$_N$ cells from C57BL/6 (Ly5.1) mice were sorted into CD5$^{lo}$Ly6C$^-$, CD5$^{hi}$Ly6C$^-$, and CD5$^{hi}$Ly6C$^+$ cells (hereafter, referred to as CD5$^{lo}$, Ly6C$^-$, and Ly6C$^+$ cells, respectively; Supplementary Fig. 1a) and were adoptively transferred to *Rag1$^{-/-}$* mice (Ly5.2) to induce T cell-mediated IBD (Fig. 1a). As expected, histological analysis of the colon on 14 d of transfer showed severe T cell-induced inflammation; however, the disease outcomes were not identical for the three subsets—the CD5$^{lo}$ subset induced the most severe destruction of colonic epithelial architecture (Fig. 1b), highest pathological scores and shortest colon length (Fig. 1c), and highest serum levels of pro-inflammatory cytokines compared with the Ly6C$^-$ and Ly6C$^+$ subsets (Fig. 1d). Moreover, even between the CD5$^{hi}$ subsets, whereas the Ly6C$^-$ subset induced a moderate level of inflammatory symptoms, the Ly6C$^+$ subset failed to do so, as judged by virtually intact colonic architecture, the least colonic inflammation and T cell infiltration, and fastest weight gain (Fig. 1b–d and Supplementary Fig. 1b–e). Together, these findings show that CD8$^+$ T$_N$ populations consist of distinct subsets with respect to their capacity to induce pathological inflammatory responses.

### CD5$^{lo}$ subset has a greater potential to induce IL-17-producing effector cells than CD5$^{hi}$ subsets

To understand the distinctly different capacities of each T$_N$ subset to induce IBD, we carefully examined early-stage functional parameters that might affect ultimate disease outcomes. The same adoptive transfer experiments, as shown in Fig. 1a, were conducted with purified T$_N$ subsets labeled with CellTrace Violet (CTV), and mesenteric lymph nodes (mLN) were analyzed on day 7 before the onset of disease symptoms (Fig. 2a). Donor cell recoveries in mLN were not different for the three T$_N$ subsets (Fig. 2b). When analyzed based on cell division kinetics, all three T$_N$ subsets exhibited similar patterns with two main

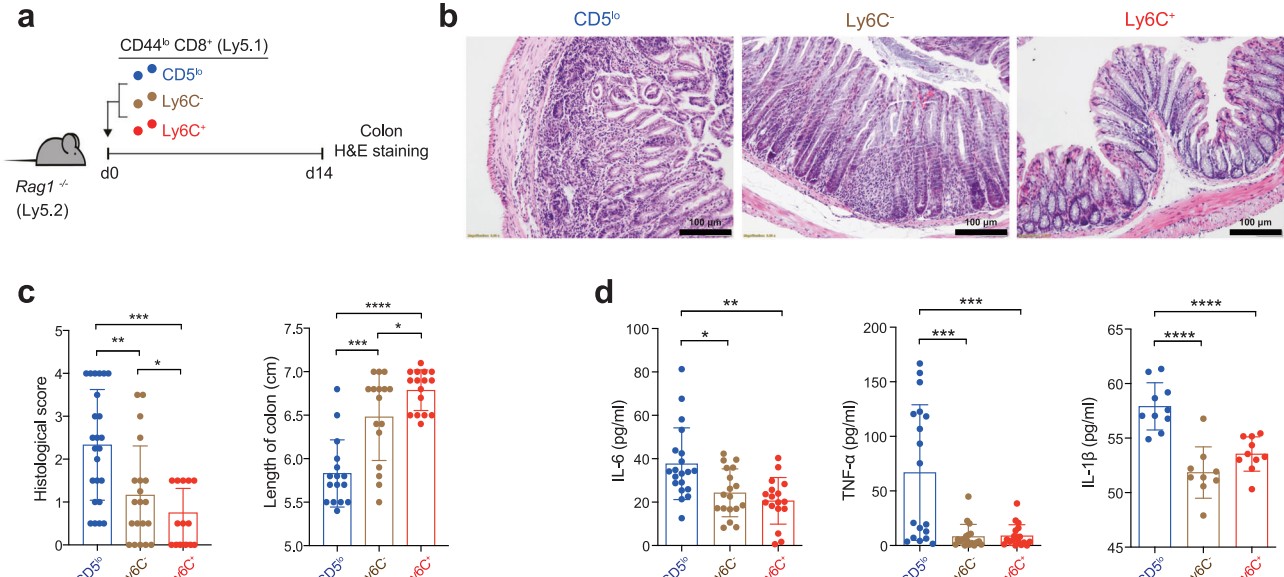

**Fig. 1 | CD8$^+$ T$_N$ subsets exhibit distinctly different capacity to induce an immunopathology in an IBD model. a** Experimental scheme for **b–d**.
**b, c** Representative H & E staining images (**b**, scale bar, 100 μm) and histopathological score (**c**, left; CD5$^{lo}$ $n = 24$, Ly6C$^-$ $n = 20$, and Ly6C$^+$ $n = 18$) and length of colon (**c**, right; $n = 16$). **d** IL-6, TNF-α, and IL-1β levels in the sera analyzed by ELISA. Data are pooled from three independent experiments (**d**, IL-6; CD5$^{lo}$ $n = 20$,

Ly6C$^-$ $n = 18$, and Ly6C$^+$ $n = 17$, TNF-α; $n = 18$, and IL-1β; CD5$^{lo}$ $n = 10$, Ly6C$^-$ $n = 9$, and Ly6C$^+$ $n = 10$) and presented as the mean ± SD (**c**) and ± SEM (**d**). Statistical significance by one-sided blind Student's $t$ test (**c**) and two-way ANOVA Multiple comparisons (**d**). $^*P < 0.05$; $^{**}P < 0.01$; $^{***}P < 0.001$; $^{****}P < 0.0001$. Source data and exact $P$ value are provided as a Source Data file.

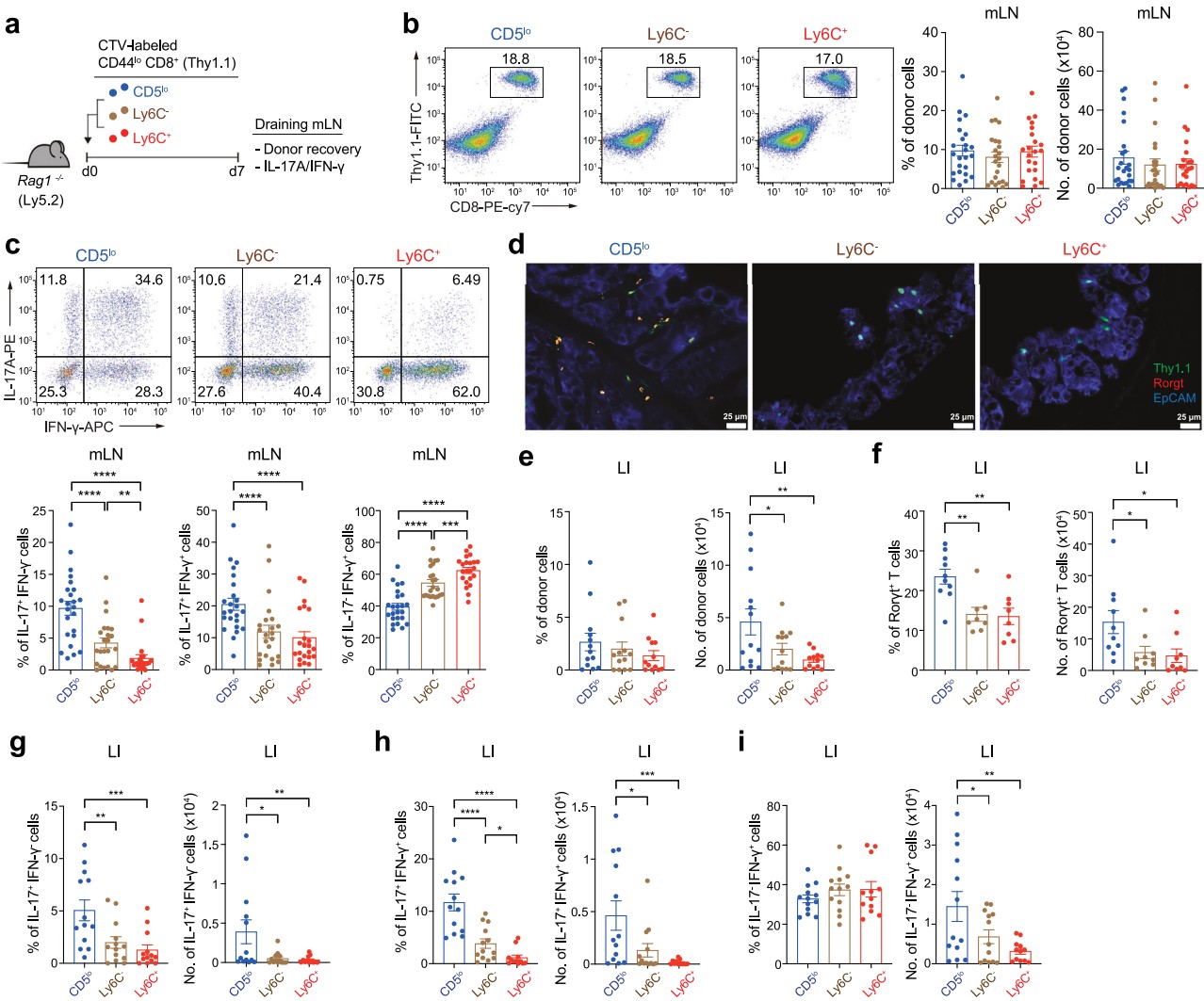

**Fig. 2 | CD5$^{lo}$ subset show greater ability to differentiate into IL-17-producing effector cells than CD5$^{hi}$ subset. a** Experimental scheme for **b–i. b** FACS plots showing Thy1.1$^+$ donor subsets (left) and donor cell recovery (right) in mLN at day 7 after transfer (CD5$^{lo}$ $n$ = 24, Ly6C$^-$ $n$ = 23, and Ly6C$^+$ $n$ = 22). **c** FACS plots for IL-17A and IFN-γ production (top) and the percentage of IL-17A$^+$IFN-γ$^-$, IL-17A$^+$IFN-γ$^+$, and IL-17A$^-$ IFN-γ$^+$ cells (bottom) in each donor subset from day 7 mLN (CD5$^{lo}$ $n$ = 24, Ly6C$^-$ $n$ = 23, and Ly6C$^+$ $n$ = 22). **d–i** Immunofluorescence images for Rorγt$^+$Thy1.1$^+$ donor cells (**d**, scale bar, 25 μm), the percentage (left) and number (right) of donor

(**e**, CD5$^{lo}$, Ly6C$^-$ $n$ = 13, and Ly6C$^+$ $n$ = 12), Rorγt$^+$ (**f**, CD5$^{lo}$ $n$ = 10, Ly6C$^-$ $n$ = 8, and Ly6C$^+$ $n$ = 8), IL-17A$^+$IFN-γ$^-$ (**g**, $n$ = 13), IL-17A$^+$IFN-γ$^+$ (**h**, CD5$^{lo}$, Ly6C$^-$ $n$ = 13, and Ly6C$^+$ $n$ = 12), and IL-17A$^-$ IFN-γ$^+$ (**i**, CD5$^{lo}$, Ly6C$^-$ $n$ = 13, and Ly6C$^+$ $n$ = 12) cells from day 7 LI. Data are pooled from three independent experiments and presented as the mean ± SEM (**b**, **c**, **e–i**). Statistical significance by two-way ANOVA Multiple comparisons $^*P < 0.05$; $^{**}P < 0.01$; $^{***}P < 0.001$; $^{****}P < 0.0001$. Source data and exact $P$ value are provided as a Source Data file.

forms of proliferation (Supplementary Fig. 2a, left): fast proliferation driven by commensal antigens and slow proliferation driven by both self antigens and homeostatic cytokines such as IL-7[7,13]. The relative frequency of these proliferations, especially slow proliferation, moderately varied among these T$_N$ subsets. The CD5$^{hi}$ subsets were more active and showed a higher frequency than the CD5$^{lo}$ subset, attributed to their higher sensitivity to cytokines, as previously reported[7]. Additionally, we assessed the in vitro proliferation capacity after TCR stimulation and found that the activation and proliferation capacities of these three T$_N$ subsets were similar, as indicated by comparable levels of activation marker expression and CTV dilution (Supplementary Fig. 2b, c). These data suggest that the different capacities of three T$_N$ subsets to induce IBD (Fig. 1b–d) were not due to differences in their early-stage T cell activation and proliferative responses.

We next compared the production of effector cytokines, IFN-γ and IL-17A, in the three donor T$_N$ subsets from day 7 mLN after short-term in vitro restimulation with PMA/ionomycin. These cells showed striking differences in their ability to produce either IL-17A or IFN-γ

(Fig. 2c, top). The proportions of IL-17A$^+$IFN-γ$^-$ and IL-17A$^+$IFN-γ$^+$ cells were the highest for the CD5$^{lo}$ subset, intermediate for the Ly6C$^-$ subset, and the lowest for the Ly6C$^+$ subset (Fig. 2c, bottom left and middle); the reverse was true for the proportions of IL-17A$^-$IFN-γ$^+$ cells, with the lowest for the CD5$^{lo}$ subset, intermediate for the Ly6C$^-$ subset, and highest for the Ly6C$^+$ subset (Fig. 2c, bottom right). When compared separately depending on their cell division profiles, nearly all IL-17A producers were detected in the fast but not slow proliferating donor populations[13], although IFN-γ producers were detected in both populations (Supplementary Fig. 2a, right), and a similar preference toward either of these cytokines was observed among the three T$_N$ donor subsets (Supplementary Fig. 2d).

Further histological analyses of the day 7 large intestine (LI) from adoptively transferred *Rag1$^{-/-}$* hosts revealed higher colonic infiltration, which was the RORγt$^+$ phenotype, and cell recovery of the CD5$^{lo}$ subset than the Ly6C$^-$ and Ly6C$^+$ subsets (Fig. 2d, e). The percentages and absolute numbers of RORγt$^+$, IL-17A$^+$IFN-γ$^-$, and IL-17A$^+$IFN-γ$^+$ cells were also higher in the CD5$^{lo}$ subset than in the Ly6C$^-$ and Ly6C$^+$

subsets (Fig. 2f–h). Notably, despite the slightly decreasing trends in the percentages of IL-17A⁻IFN-γ⁺ cells, their absolute cell numbers were much higher in the CD5$^{lo}$ donor subset than in either of the CD5$^{hi}$ subsets (Fig. 2i), owing to the enhanced cell recovery of the CD5$^{lo}$ subset in the colon (Fig. 2e). Consistent with the data on day 7, massive colon infiltration and enhanced recovery of CD5$^{lo}$ donor cells were clearly detected on day 14 (Supplementary Fig. 2e, f), with much higher proportions of IL-17A⁺IFN-γ⁻, IL-17A⁺IFN-γ⁺, and RORγt⁺ cells and lower proportions of IL-17A⁻IFN-γ⁺ cells in the CD5$^{lo}$ subset than in the Ly6C⁻ and Ly6C⁺ subsets (Supplementary Fig. 2g–j).

To further understand how the increased IL-17 production by the CD5$^{lo}$ subset is associated with its ability to induce rapid and robust IBD, we administered BrdU via drinking water (to measure proliferative activity) or orally administered FITC-conjugated dextran (to measure gut epithelial leakage) to *Rag1*⁻/⁻ mice adoptively transferred with CD5$^{lo}$, Ly6C⁻, and Ly6C⁺ subsets 48 or 4 h prior to analysis on days 7 and 14 post-transfer (Supplementary Fig. 2k, l). The CD5$^{lo}$ subset, compared to Ly6C⁻ and Ly6C⁺ subsets, showed significantly enhanced proliferative responses and augmented gut epithelial barrier leakages, as evidenced by the higher proportion of BrdU⁺ donor cells in the colon (Supplementary Fig. 2k) and the level of FITC-dextran in the serum (Supplementary Fig. 2l) at day 14, albeit comparable at day 7, post-transfer. These data strongly suggest that the CD5$^{lo}$ subset (as IL-17-producing cells) is able to reach the colon more quickly and efficiently, induce gut epithelial damage, release more gut-derived antigens, and gradually increase in sufficient numbers via robust antigen-dependent proliferative responses, perhaps between 7 and 14 days post-transfer. Together, these findings demonstrate that CD8⁺ T$_N$ cells are heterogeneous in differentiating into IL-17A-producing effector cells and these different abilities are associated with the distinctly different robustness of IBD outcomes observed in CD5$^{lo}$, Ly6C⁻, and Ly6C⁺ subsets.

## The fast and robust IBD induced by CD5$^{lo}$ subset is CCR6 dependent

The significant increase in the migration ability of CD5$^{lo}$ cells from the mLN to the LI, coupled with the elevated number of colonic CD5$^{lo}$ cells and the severity of IBD, implies a potential difference in the expression of gut-homing receptors on these cells. In fact, we found enhanced levels of CCR5, CCR6, and GPR15 (but not α4β7)—chemoattractant receptors that direct T cell migration to the colon[17–20]—in the CD5$^{lo}$ subset compared with that in the Ly6C⁻ and Ly6C⁺ subsets obtained from day 7 mLN of *Rag1*⁻/⁻ mice after the transfer (Fig. 3a and Supplementary Fig. 3a). Furthermore, upon isolating the two most distinct subsets, CD5$^{lo}$ and Ly6C⁺, from *Rag1*⁻/⁻ mice at day 7 post-transfer and culturing them in transwell plates with CCL4 and CCL20 (ligands for CCR5 and CCR6, respectively), it became evident that CD5$^{lo}$ cells exhibited a higher migratory ability compared to Ly6C⁺ cells (Supplementary Fig. 3b, c). These findings suggest that the robust colonic infiltration and induction of IBD driven by the CD5$^{lo}$ subset may be attributed, at least in part, to its enhanced gut-homing ability through augmented interactions with chemokine receptors and ligands. To further understand the role of chemokine ligand/receptor interactions, particularly via CCR6, we performed adoptive transfer experiments in *Rag1*⁻/⁻ mice with CD5$^{lo}$, Ly6C⁻, and Ly6C⁺ CD8⁺ T$_N$ subsets purified from wild type (WT; *Ccr6*⁺/⁺) or *Ccr6*⁻/⁻ mice. While the WT CD5$^{lo}$ subset induced the most severe colonic disruption and infiltration than did the WT Ly6C⁻ and Ly6C⁺ subsets on day 14, the *Ccr6*⁻/⁻ CD5$^{lo}$ subset failed to do so, with near intact colon architecture and poor infiltration similar to that seen in the *Ccr6*⁻/⁻ Ly6C⁻ and Ly6C⁺ subsets (Fig. 3b–d).

Notably, despite such a defect in inducing IBD, the ability of the *Ccr6*⁻/⁻ CD5$^{lo}$ subset to produce IL-17 was still remarkably high, as evidenced by the enhanced frequencies (albeit not numbers) of IL-17A-producing cells in the *Ccr6*⁻/⁻ CD5$^{lo}$ subset compared with that in the

*Ccr6*⁻/⁻ Ly6C⁻ and Ly6C⁺ subsets in the day 14 LI (Fig. 3e and Supplementary Fig. 3d–f) and mLN (Supplementary Fig. 3g–j). Consistent with the enhanced IL-17A-producing ability, the *Ccr6*⁻/⁻ CD5$^{lo}$ subset was able to induce a delayed (detectable at day 21), but the highest, onset of IBD symptoms (Fig. 3f, g), as well as the highest frequencies and numbers of IL-17A⁺ cells compared with those induced by the *Ccr6*⁻/⁻ Ly6C⁻ and Ly6C⁺ subsets (both in the LI and mLN; Fig. 3h and Supplementary Fig. 3k–n). In addition to the role of CCR6, it is important to note that the above results do not preclude the involvement of other chemokine receptors, particularly CCR5 and GPR15, as several studies using similar IBD models have emphasized the significance of these gut-homing receptors[17,20]. Hence, these findings strongly support our notion that, under inflammatory condition, the CD5$^{lo}$ CD8⁺ T$_N$ subset is highly predisposed to differentiate into IL-17 producers, promoting the augmented expression of the gut-homing receptors, thereby rendering the CD5$^{lo}$ subset more quickly migrate to the colon and trigger a fast and robust colonic inflammation over the CD5$^{hi}$ subsets (particularly the Ly6C⁺ subset).

## CD5$^{lo}$ subset exhibits a higher pathogenic potential than CD5$^{hi}$ subsets in a GVHD condition

To further corroborate the notion that the CD5$^{lo}$ CD8⁺ T$_N$ subset is strongly poised to differentiate toward IL-17 producers that become pathogenic under inflammatory conditions, we extended our analysis to other inflammatory disease model, GVHD. Lethally irradiated BALB/c mice (Ly5.2) were reconstituted by adoptive transfer with allogeneic B6-derived bone marrow cells (Ly5.1) in the absence or presence of purified CD5$^{lo}$, Ly6C⁻, or Ly6C⁺ subsets (Thy1.1) (Fig. 4a). Monitoring the BALB/c recipients revealed that the co-transfer of allogeneic B6 CD8⁺ T$_N$ subsets induced GVHD; however, the disease severity was distinct among each subset, with CD5$^{lo}$ cells being worse than Ly6C⁻ and Ly6C⁺ cells, as evidenced by the maximum loss in body weight and the least survival during the onset of GVHD (Fig. 4b, c). Consistent with this, tissue inflammation and infiltration into the kidney and LI of the day 15 BALB/c recipients were apparently the highest for the CD5$^{lo}$ subset, intermediate for the Ly6C⁻ subset, and lowest for the Ly6C⁺ subset (Fig. 4d). Furthermore, these inflammatory symptoms were associated with higher frequencies and numbers of IL-17A⁺ (both GM-CSF⁻ and GM-CSF⁺) cells and, to a lesser extent, of GM-CSF⁺ cells and IFN-γ⁺ cells in the CD5$^{lo}$ subset than in the Ly6C⁻ and Ly6C⁺ subsets (from the day 7 peripheral LN; Fig. 4e–h). These results, along with the data from the IBD model, strongly suggest that the Tc17 differentiation program is cell-intrinsic, and that it is regulated differently among the CD8⁺ T$_N$ subsets, rendering the CD5$^{lo}$ subset the most skewed toward a highly pathogenic IL-17 producer, presumably in a wide range of acute and/or chronic inflammatory disease contexts.

## High ability of the CD5$^{lo}$ subset to induce IL-17-producing cells is intrinsic and independent of its TCR diversity and specificity

To further address whether the strong Tc17 bias of the CD5$^{lo}$ CD8⁺ T$_N$ subset (and conversely, the poor Tc17 skewing of the Ly6C⁻ and Ly6C⁺ subsets) is cell-intrinsic, we adopted an in vitro culture system to rule out a potential effect of cell-extrinsic environment that might vary depending on the in vivo disease models used. Purified CD5$^{lo}$, Ly6C⁻, and Ly6C⁺ CD8⁺ T$_N$ subsets were cultured under a condition that promoted Tc17 differentiation by stimulating them with anti-CD3/CD28 in the presence of IL-6 and TGF-β (Fig. 5a). The results from the in vitro Tc17 culture were surprisingly similar to those observed in the in vivo disease settings, as illustrated by the higher proportion of IL-17A⁺ cells and the lower proportion of IFN-γ⁺ cells, as well as by the higher proportion of Rorγt⁺ cells and the lower proportion of Eomes⁺ cells in the CD5$^{lo}$ subset than in the Ly6C⁻ and Ly6C⁺ subsets (Fig. 5b–d). Enhanced IL-17A production in the CD5$^{lo}$ subset was also observed using different combinations and concentrations of various Tc17-polarizing cytokines, such as IL-6, TGF-β, IL-1β, IL-21, and IL-23

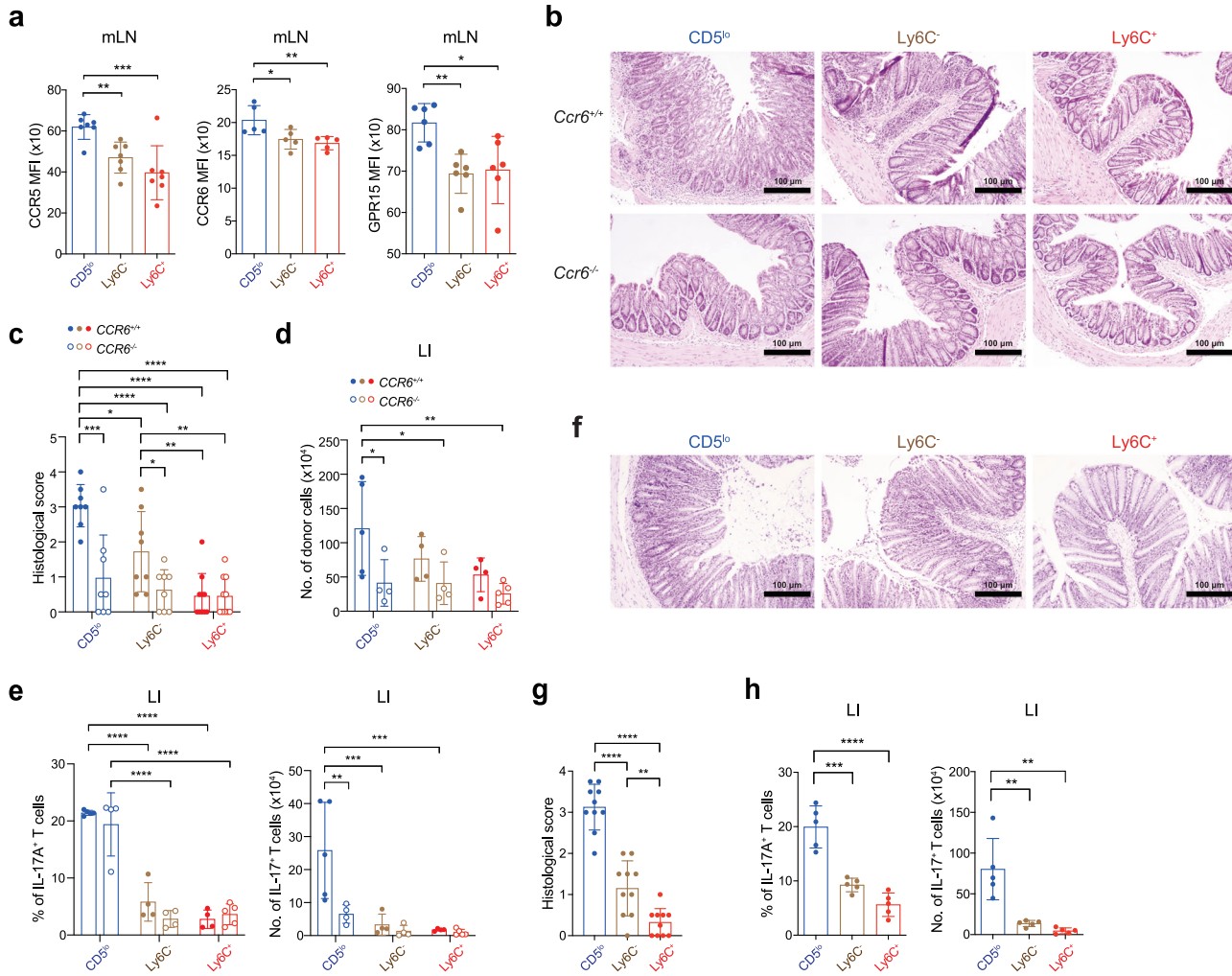

**Fig. 3 | CD5^lo subset show enhanced ability to migrate to the colon in a CCR6 dependent manner. a** Expression levels (as the mean fluorescence intensity, MFI) of CCR5, CCR6, and GPR15 of transferred donor subsets from day 7 mLN (CCR5 $n = 7$, CCR6 $n = 5$, and GPR15 $n = 6$ per group). **b–e** Representative H&E staining images (**b**, scale bar, 100 μm), histopathological score (**c**, CD5^lo $n = 4$, Ly6C^- $n = 4$, and Ly6C^+ $n = 5$), the number of donor cells (**d**, $Ccr6^{+/+}$; CD5^lo $n = 5$, Ly6C^- $n = 4$, and Ly6C^+ $n = 4$, and $Ccr6^{-/-}$; CD5^lo $n = 4$, Ly6C^- $n = 4$, and Ly6C^+ $n = 5$), and the percentage (left) and number (right) of IL-17A^+ cells (**e**, $Ccr6^{+/+}$; CD5^lo $n = 5$, Ly6C^- $n = 4$, and Ly6C^+ $n = 4$, and $Ccr6^{-/-}$; CD5^lo $n = 4$, Ly6C^- $n = 4$, and Ly6C^+ $n = 5$) analyzed in LI at day 14 after adoptive transfer with either $Ccr6^{+/+}$ or $Ccr6^{-/-}$ CD8^+ T_N subsets. **f–h** Representative H&E staining images (**f**, scale bar, 100 μm), histopathological score (**g**, $n = 10$), and the percentage (left) and number (right) of IL-17A^+ donor cells (**h**, $n = 5$) analyzed in LI at day 21 after transfer with $Ccr6^{-/-}$ CD8^+ T_N subsets. Data are representative of two independent experiments and presented as the mean ± SD (**a**, **c–e**, **g**, **h**). Statistical significance by one-sided blind Student's $t$ test (**c**, **g**) and two-way ANOVA Multiple comparisons (**a**, **d**, **e**, **g**, **h**). *, $P < 0.05$; **, $P < 0.01$; ***, $P < 0.001$; ****, $P < 0.0001$. Source data and exact $P$ value are provided as a Source Data file.

(Supplementary Fig. 4a, b), implying a cell-intrinsic, but not a cell-extrinsic, phenomenon.

To validate whether the observed cell-intrinsic property was not restricted to the B6 CD8^+ T_N subsets bearing polyclonal TCR repertoire, we performed the above in vitro Tc17 culture experiments with CD5^lo, Ly6C^-, and Ly6C^+ CD8^+ T_N subsets purified from P14 TCR transgenic mice[5]. Similar to the results for the B6 CD8^+ T_N subsets, the P14 CD5^lo subset exhibited a higher proportion of IL-17A^+ cells and a lower proportion of IFN-γ^+ cells than did the P14 Ly6C^- and Ly6C^+ subsets after Tc17 culture, either with anti-CD3/CD28 (Fig. 5e) or dendritic cells pulsed with gp33 peptide (a cognate antigen for P14 TCR; Supplementary Fig. 4c). Furthermore, when analyzed using bulk RNA-seq at day 3 after Tc17 culture, compared with Ly6C^- and Ly6C^+ subsets, the CD5^lo subset exhibited a higher enrichment of publicly available Th17 signature gene sets (GSE14026) and a lower enrichment of Th1 signature gene sets (GSE15930) (Fig. 5f). These data were also confirmed by qRT-PCR (i.e., a higher *Il17a*, *Rorc*, and *Irf4* expression and a lower *Ifng*, *Eomes* and *Tbx21* expression; Supplementary Fig. 4d).

Together, these data further support the notion that the CD5^lo subset, relative to its CD5^hi counterparts, is strongly poised to undergo the Tc17 differentiation program that is acquired in a cell-intrinsic manner regardless of its TCR diversity and specificity.

## High Tc17-skewing ability of the CD5^lo subset is independent of its responsiveness to cytokines that regulate Tc17 differentiation

To understand the mechanisms underlying differential cell-intrinsic property for Tc17 differentiation, we examined the expression levels of receptors for Tc17-polarizing cytokines, such as IL-6Rα, GP130, TGFβR1, and TGFβR2 (for IL-6 and TGF-β). Notably, the expression of these receptors was not increased but was instead significantly decreased in the CD5^lo subset compared with that in the Ly6C^- and Ly6C^+ subsets (Supplementary Fig. 4e). Consistent with the reduced receptor expression, the CD5^lo subset exhibited a much lower activation of STAT3 and SMAD2/3, relative to the Ly6C^- and Ly6C^+ subsets, upon treatment with IL-6 and TGF-β, respectively (Supplementary

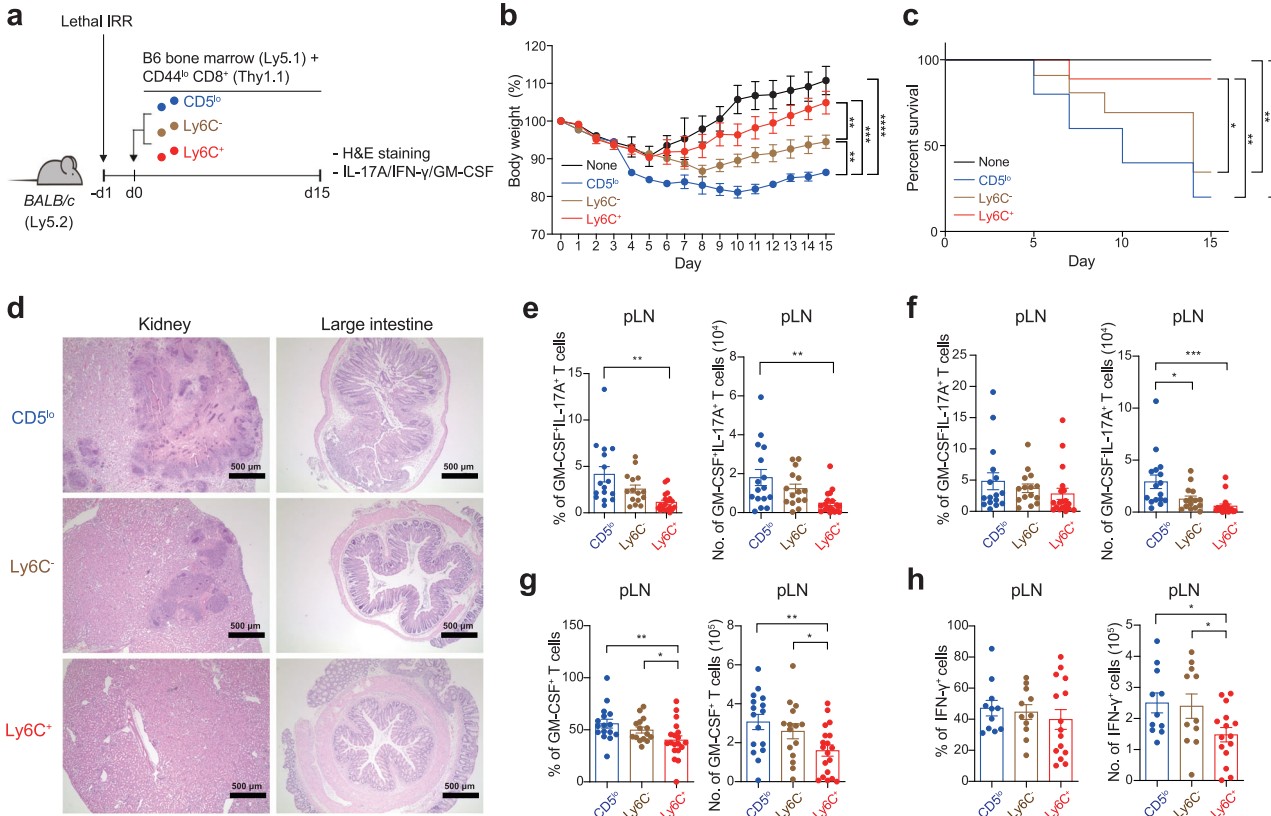

**Fig. 4 | CD5^lo subset exhibits higher pathogenic potential than CD5^hi subsets in a GVHD condition. a** Experimental scheme for **b–h. b, c** Body weight changes (**b**, None $n = 5$, CD5^lo $n = 6$, Ly6C^− $n = 5$, and Ly6C^+ $n = 6$) and survival rate (**c**, None $n = 5$, CD5^lo $n = 6$, Ly6C^− $n = 5$, and Ly6C^+ $n = 6$) of BALB/c recipient mice after adoptive co-transfer with C57BL/6 bone marrow cells and each of CD8^+ T_N subsets. **d–h** Representative H & E staining images analyzed in kidney and LI at day 15 after transfer (**d**, scale bar, 500 μm), and the percentage and number of GM-CSF^−IL-17A^+ (**e**, left; CD5^lo $n = 16$, Ly6C^− $n = 15$, and Ly6C^+ $n = 17$, and right; CD5^lo $n = 16$, Ly6C^− $n = 15$, and Ly6C^+ $n = 19$), GM-CSF^− IL-17A^+ (**f**, CD5^lo $n = 16$, Ly6C^− $n = 15$, and

Ly6C^+ $n = 19$), GM-CSF^+ (**g**, CD5^lo $n = 16$, Ly6C^− $n = 15$, and Ly6C^+ $n = 19$), and IFN-γ^+ (**h**, left; CD5^lo $n = 16$, Ly6C^− $n = 15$, and Ly6C^+ $n = 19$, and right; CD5^lo $n = 10$, Ly6C^− $n = 11$, and Ly6C^+ $n = 15$) donor cells analyzed in LN at day 7 after transfer. Data are representative of three independent experiments or pooled from three independent experiments and presented as the mean ± SEM (**b, e–h**). Statistical significance by two-tailed Mann–Whitney test (**b**), Log-rank (Mantel-Cox) test (**c**) and two-way ANOVA Multiple comparisons (**e–h**). *$P < 0.05$; **$P < 0.01$; ***$P < 0.001$; ****$P < 0.0001$. Source data and exact $P$ value are provided as a Source Data file.

Fig. 4f). These data suggest that the higher Tc17-skewing property of the CD5^lo subset is not due to enhanced expression of the receptor and/or downstream signaling for Tc17-polarizing cytokines.

Having observed a more intense−despite being poor Tc17 differentiation−cytokine signaling (both phospho (p)-STAT3 and p-SMAD2/3 upon IL-6 and TGF-β treatment) in Ly6C^− and to a greater extent Ly6C^+ subsets (Supplementary Fig. 4f), we reasoned that there is an inhibitory mechanism that might strongly antagonize the CD5^hi subsets to undergo the Tc17 program. As IL-2 negatively regulates both Th17 and Tc17 differentiation[15,21], we examined whether the poor Tc17-skewing property of the CD5^hi subsets (particularly of the Ly6C^+ cells) simply reflects more intense IL-2 signaling and, conversely, whether the strong Tc17 bias of the CD5^lo subset results from weaker IL-2 signaling. Indeed, the production of IL-17 by CD8^+ T_N subsets cultured under Tc17-polarizing conditions decreased upon IL-2 (recombinant murine (rm) IL-2; Supplementary Fig. 4g) addition but increased substantially either by blocking IL-2 (using anti-IL-2 and anti-IL-2Rβ; Fig. 5g, h) or by utilizing cells lacking IL-2Rβ (Supplementary Fig. 4h). In contrast, the production of IFN-γ further increased and decreased by the addition and blocking of IL-2, respectively (Fig. 5g, h and Supplementary Fig. 4g, h). Of note, despite the overall enhancement in the proportion of IL-17A^+ cells in all the CD8^+ T_N subsets by IL-2 blockade, the CD5^lo subset still exhibited a higher production of IL-17 (Fig. 5g, h and Supplementary Fig. 4h) and greater expression of Tc17-associated signature genes than did the Ly6C^− and Ly6C^+ subsets (Fig. 5i),

implying the existence of an IL-2-independent regulatory mechanism for skewed Tc17 differentiation.

## CD5^lo subset exhibits greater TCR responsiveness with augmented IRF4 expression

Given that a strong TCR signal is a stimulator of Th17 differentiation[22,23], we investigated the effect of increasing the strength of TCR signal, especially in the presence of excess amounts of exogenous IL-2, to avoid a possible confounding effect of endogenously produced IL-2 at different levels during stimulation. Overall, increasing the amount of anti-CD3 under Tc17 culture resulted in a substantial increase in the proportion of IL-17A^+ cells, but conversely decreased the proportion of IFN-γ^+ cells in all the CD8^+ T_N subsets (cultured in the presence of either recombinant human (rh) or rmIL-2; Fig. 5j and Supplementary Fig. 4i); however, even under these conditions, the higher IL-17- (but lower IFN-γ-) producing ability of the CD5^lo subset compared with those of the Ly6C^− and Ly6C^+ subsets remained unchanged.

We next investigated whether the differential Tc17-skewing property is related to the difference in the sensitivity of TCR signaling among the CD8^+ T_N subsets. In fact, the CD5^lo subset was more responsive to TCR ligation, as evidenced by the higher proportions and amounts of p-ERK in the CD5^lo subset than in the Ly6C^− and Ly6C^+ subsets after treatment with anti-CD3 (Supplementary Fig. 5a, b). Moreover, consistent with the enhanced TCR responsiveness, the level

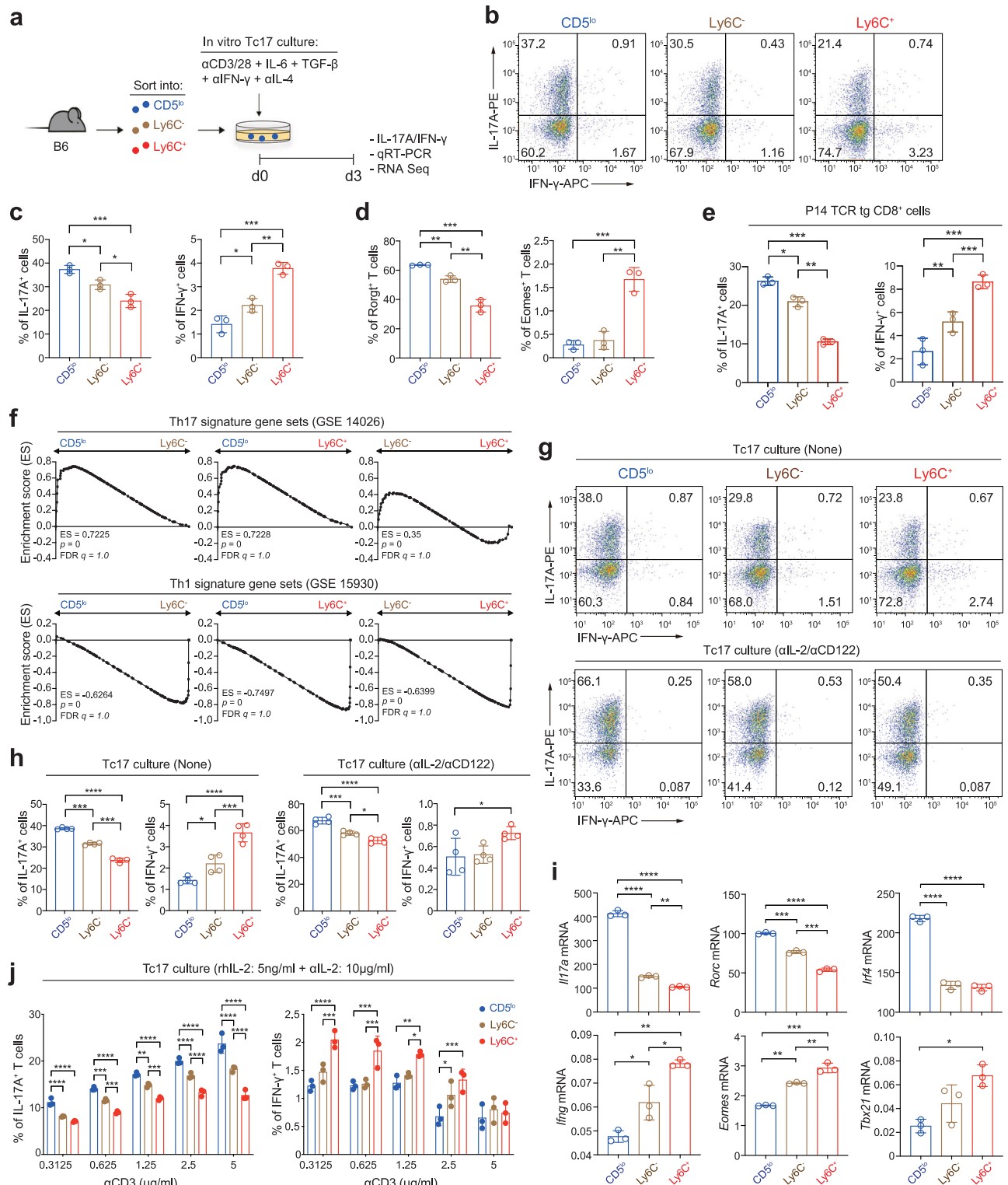

**Fig. 5 | Enhanced Tc17 differentiation of CD5$^{lo}$ subset is a cell-intrinsic property.** **a** Experimental scheme for **b–f**. **b–d** Representative FACS plots for IL-17A/IFN-γ production (**b**), and the percentage of IL-17A$^+$ and IFN-γ$^+$ cells (**c**) and Rorγt$^+$ and Eomes$^+$ cells (**d**) after in vitro culture of each CD8$^+$ T$_N$ subset under Tc17 polarization condition. **e** Percentage of IL-17A$^+$ and IFN-γ$^+$ cells after Tc17-polarizing culture with P14 CD8$^+$ T$_N$ subsets. **f** GSEA of Tc17-polarized B6 CD8$^+$ T$_N$ subsets for Th17 (top) and Th1 signature gene sets (bottom). **g–i** Representative FACS plots for IL-17A/IFN-γ production (**g**), the percentage of IL-17A$^+$ and IFN-γ$^+$ cells (**h**), and qRT-

PCR data for *Il17a, Rorc, Irf4, Ifng, Eomes,* and *Tbx21* (**i**) after Tc17-polarizing culture with B6 CD8$^+$ T$_N$ subsets in the presence (**g–i**) or absence (**g, h**) of anti-IL-2/CD122. **j,** Percentage of IL-17A$^+$ and IFN-γ$^+$ cells after Tc17-polarizing culture with B6 CD8$^+$ T$_N$ subsets in the presence of rhIL-2 and anti-mIL-2. Data are representative of two (**i**, $n = 3$ per experiment) and three (**b–e, g, h, j**, $n = 3$ per experiment) independent experiments and presented as the mean ± SD (**c–e, h–j**). Statistical significance by two-way ANOVA Multiple comparisons. $^*P < 0.05$; $^{**}P < 0.01$; $^{***}P < 0.001$; $^{****}P < 0.0001$. Source data and exact $P$ value are provided as a Source Data file.

of interferon regulatory factor 4 (IRF4), a key transcription factor for inducing the Th17 and Tc17 program and its expression is directly dependent on the strength of TCR signaling[24–26], was markedly increased in the CD5[lo] subset compared with that in the Ly6C[−] and Ly6C[+] subsets after TCR stimulation (Fig. 6a). Most importantly, our chromatin immunoprecipitation (ChIP) analysis (conducted at 24 and 72 h after Tc17 culture) revealed a much higher occupancy of IRF4 bound to *Rorc* and *Il17a* gene locus in the CD5[lo] subset than in the Ly6C[−] and Ly6C[+] subsets (Fig. 6b and Supplementary Fig. 5c). These results suggest that the CD5[lo] subset more effectively activates the Tc17 program because of its higher cell-intrinsic TCR responsiveness and, accordingly, greater IRF4 expression.

## Levels of endogenous SMAD3 differ among CD8[+] $T_N$ subsets and regulate their Tc17 differentiation potential

Although the relatively higher TCR signaling and IRF4 expression is a strong basis for the enhanced Tc17-skewing property of the CD5[lo] subset, the reason for the poor ability of the CD5[hi] subsets to become IL-17 producers even in the presence of strong TCR signals and/or concurrent IL-2 blockade remains unclear. We sought to explore the mechanism behind the poor Tc17-skewing property of the CD5[hi] subsets, particularly of the Ly6C[+] subset, and found a crucial role for SMAD3 in this phenomenon. Notably, the basal expression of SMAD3 was not identical among the freshly isolated ex vivo CD8[+] $T_N$ subsets, with the lowest for the CD5[lo] subset, intermediate for the Ly6C[−] subset, and highest for the Ly6C[+] subset (Fig. 6c); similar results were observed for p-SMAD3, implying continuous but differential levels of tonic TGF-β signal in vivo. Likewise, the expression of SMAD2 (and p-SMAD2) was also highly upregulated in the Ly6C[+] subset compared with that in the CD5[lo] and Ly6C[−] subsets (Supplementary Fig. 5d).

As SMAD3, but not SMAD2, negatively regulates Th17 differentiation[27,28], we further investigated the impact of different levels of SMAD3 on the Tc17-skewing potential. Notably, in line with the higher SMAD3 levels in the Ly6C[+] subset, ChIP analysis showed much greater SMAD3 occupancy in the *Rorc* and *Il17a* gene promoter regions in the Ly6C[+] subset than in the CD5[lo] and Ly6C[−] subsets (analyzed at 24 and 72 h after Tc17 culture; Fig. 6d and Supplementary Fig. 5e), which was in stark contrast to the IRF4 ChIP data (Fig. 6b). Most importantly, the forced overexpression of SMAD3 by retroviral transduction encoding *SMAD3* gene (~2–3-fold higher expression than for the empty vector control; Supplementary Fig. 5f) led to a marked suppression in the production of IL-17A in nearly all the Tc17-cultured CD8[+] $T_N$ subsets, with the CD5[lo] subset exerting the most profound effect (comparing untransduced GFP[−] vs. transduced GFP[+] cells; Fig. 6e); notably, there were no such differences in cells transduced with the empty control vector (Supplementary Fig. 5g). In a sharp contrast, forced downregulation of SMAD3 by retroviral transduction encoding *SMAD3* shRNA (~1.5-fold lower expression than for empty vector control; Supplementary Fig. 5h) significantly enhanced the production of IL-17A in all the Tc17-cultured CD8[+] $T_N$ subsets, with a prominent effect on the CD5[hi] subsets, particularly on the Ly6C[+] cells (Fig. 6f); no effects were observed in cells transduced with the empty control vector (Supplementary Fig. 5i).

Providing further support of the inhibitory role of SMAD3, adoptive transfer of activated total CD8[+] $T_N$ cells, retrovirally transduced either for SMAD3 overexpression or downregulation (Fig. 6g, h), into *Rag1[−/−]* mice showed a decrease or increase in the proportion of IL-17A[+] cells, respectively, when analyzed for 14 d colon (Fig. 6i). The production of IFN-γ was also moderately affected by retrovirally altered SMAD3 expression; however, unlike for the production of IL-17A, the opposite effects were generally observed, that is, increase and decrease in the proportion of IFN-γ[+] cells upon overexpression and downregulation of SMAD3, respectively (Fig. 6e, f, j). Together, these results highlight the importance of different levels of endogenous SMAD3, which acts as a strong negative regulator for inducing the Tc17

program, supportive of the idea that the Ly6C[+] subset, among the CD8[+] $T_N$ subsets, has the least capacity to become Tc17 cells, owing to its higher basal amounts of SMAD3.

## Developmental self-reactivity of CD8[+] $T_N$ cells positively correlates with basal SMAD3 levels but negatively with Tc17 differentiation potential

To investigate whether peripheral CD8[+] $T_N$ subsets developmentally acquire differential SMAD3 expressions, we analyzed thymocytes. Notably, for CD4[−]CD8[+] single-positive (SP) mature (CD24[lo]) thymocytes, the expression of SMAD3 was higher in the CD5[hi] subsets (both the Ly6C[−] and, to a greater extent, the Ly6C[+] subsets) than in the CD5[lo] subset (Fig. 7a). Moreover, similar to the observations in the peripheral $T_N$ subsets, stimulation of these thymic $T_N$ subsets under Tc17 culture conditions (Fig. 7b) resulted in a strong inverse correlation between the SMAD3 levels and IL-17A production, as evidenced by the least proportion of IL-17A[+] cells for the Ly6C[+] subset, intermediate for the Ly6C[−] subset, and highest for the CD5[lo] subset (Fig. 7c, d), and a reverse scenario for IFN-γ production.

We further validated a possible relationship between developmental self-reactivity (i.e., CD5 level), SMAD3 expression, and Tc17 differentiation potential by comparing various TCR transgenic CD8[+] $T_N$ cells, such as HY, Pmel-1, P14, and OT-I. Stimulation of these monoclonal CD8[+] $T_N$ cells in Tc17 culture induced the IL-17-producing ability in the following hierarchical order: HY > Pmel-1 > B6 (polyclonal) > P14 > OT-I (Fig. 7e). This order in the production of IL-17A among these $T_N$ populations was inversely correlated with their intrinsic TCR affinity for self-ligands (judged by the CD5 levels: Fig. 7f, left) and, most importantly, with the basal SMAD3 levels (Fig. 7f, right); these two parameters, CD5 and SMAD3, showed the strong positive relationship (Fig. 7g). Similar results were obtained for the expression of Tc17-associated signature genes by further in-depth comparisons of HY and OT-I (Supplementary Fig. 6a, b).

To further corroborate whether the aforementioned relationship also applies to humans, we analyzed human peripheral blood $T_N$ (CD3[+]CCR7[+]CD45RA[+]CD95[−]) cells from healthy individuals. The levels of CD5 on human CD8[+] (and even CD4[+]) $T_N$ cells were not identical, but were rather different among individuals, and surprisingly, were positively correlated with the endogenous levels of SMAD3 (Fig. 7h and Supplementary Fig. 6c), which was consistent with the observations in CD8[+] $T_N$ cells from different TCR transgenic mice (Fig. 7g). Furthermore, after Tc17 culture, human CD8[+] (and CD4[+]) $T_N$ cells from different individuals produced variable levels of IL-17 (~2–15% IL-17A[+] cells), and the production of IL-17 was inversely correlated with the levels of both CD5 and SMAD3 (Fig. 7i and Supplementary Fig. 6d). Based on the close similarity between mice and humans, we also directly compared the Tc17 differentiation potential between the CD5[lo] and CD5[hi] subsets purified from each individual human CD8[+] $T_N$ population. The results from the Tc17 culture revealed a moderate, but significantly higher, proportion of IL-17A[+] cells in the CD5[lo] subset than in the CD5[hi] subset, although IFN-γ[+] cells were not different (Fig. 7j).

As Tc17 cells are involved in the pathogenesis of several inflammatory diseases in humans[11,12,14,15], we sought to determine whether the strong inverse relationship between Tc17 differentiation and SMAD3 expression applies in actual clinical settings, such as in ulcerative colitis (UC). By analyzing publicly available scRNA-seq datasets from patients with UC (GSE162335), we found that ~18% of the total CD3[+] cells expressed either of two transcripts, *IL17A* or *SMAD3*, whereas the remaining 82% of the cells did not express both (Supplementary Fig. 6e). When further analyzed with cells expressing either *IL17A* or *SMAD3* transcripts, we observed a striking inverse correlation in the expression of *IL17A* and *SMAD3*, as indicated by clear non-overlapping clusters of *IL17A*- and *SMAD3*-expressing cells (Fig. 7k). This distinction (i.e., the expression of *IL17A* being completely opposite to that of

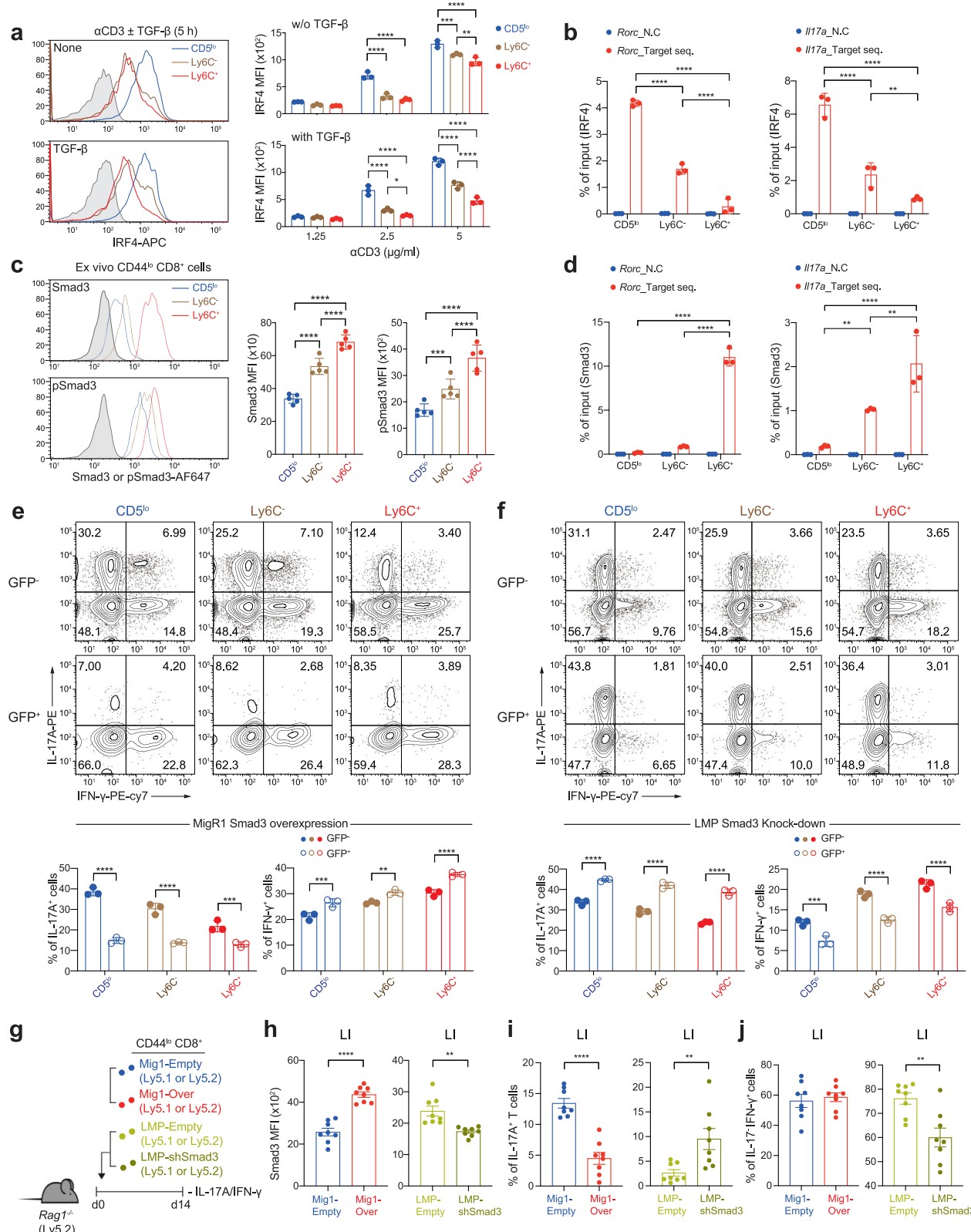

*SMAD3*) was also evident in both CD4⁺ and CD8⁺ T cells (Fig. 7l and Supplementary Fig. 6f). A similar opposite regulation between *IL17A* and *SMAD3*−rendering cells barely expressing both *IL17A* and *SMAD3* simultaneously−was also observed in different UC patient cohort (GSE116222) and other inflammatory disorders, such as Crohn's disease (GSE163314), spondyloarthritis (GSE163314), and psoriasis (GSE151177) (Supplementary Fig. 6g). It should be noted that, unlike

the observed relationship between *IL17A* and *SMAD3*, we did not observe a similar correlation with *CD5* due to the inefficiency and low resolution of measuring *CD5* transcripts based on scRNA-seq analysis. Collectively, these findings indicate that thymically acquired intrinsic self-reactivity determines the relative capacity of individual CD8⁺ $T_N$ cells to undergo a pathogenic Tc17 program by adjusting the basal levels of endogenous SMAD3 differently.

**Fig. 6 | CD5$^{lo}$ subset is poised to better differentiate toward Tc17 cells due to higher TCR responsiveness and basal SMAD3 expression. a** Histograms (left) and MFI levels (right) for IRF4 expression among CD8$^+$ T$_N$ subsets stimulated with either anti-CD3 in the presence or absence of TGF-β (left) or with various concentrations of anti-CD3 (right). **b** B6 CD8$^+$ T$_N$ subsets were activated under Tc17-polarizing conditions for 24 h and subjected to ChIP using IRF4 antibody. Eluted DNA was analyzed by qPCR. **c** Endogenous levels of SMAD3 and p-SMAD3 in ex vivo B6 CD8$^+$ T$_N$ subsets shown in histogram (left) and MFI (right) ($n = 5$ mice/group). **d** B6 CD8$^+$ T$_N$ subsets were activated under Tc17-polarizing conditions for 24 h and subjected to ChIP using SMAD3 antibody. Eluted DNA was analyzed by qPCR. **e** Representative FACS plots for IL-17A/IFN-γ production of Tc17-polarized CD8$^+$ T$_N$ subsets transduced with MigR-1 vector encoding *SMAD3* (top) and the percentage of IL-17A$^+$ (bottom left) and IFN-γ$^+$ cells (bottom right) in untransduced GFP$^-$ and transduced GFP$^+$ cells. **f** Representative FACS plots for IL-17A/IFN-γ production of Tc17-polarized CD8$^+$ T$_N$ subsets transduced with LMP vector containing *SMAD3* shRNA (top) and the percentage of IL-17A$^+$ (bottom left) and IFN-γ$^+$ cells (bottom right) in untransduced GFP$^-$ and transduced GFP$^+$ cells. **g** Experimental scheme for **h–j**. **h** Expression levels of SMAD3 in donor cells (from LI) transduced with either MigR-1 vector encoding *SMAD3* (left) or LMP vector containing *SMAD3* shRNA (right). **i, j** Percentage of IL-17A$^+$ cells (**i**) and IL-17A$^-$ IFN-γ$^+$ cells (**j**) in transduced GFP$^+$ donor cells from LI. Data are representative of three independent experiments (**a, b, d–f**, $n = 3$; **c**, $n = 5$ per experiment) or pooled from two to independent experiments (**h–j**, $n = 8$) and presented as the mean ± SD (**a–f**), and ± SEM (**h–j**). Statistical significance by two-way ANOVA Multiple comparisons. $^*P < 0.05$; $^{**}P < 0.01$; $^{***}P < 0.001$; $^{****}P < 0.0001$. Source data and exact $P$ value are provided as a Source Data file.

## Discussion

Acquisition of the cytotoxic effector functions of CD8$^+$ T cells requires sophisticated regulation of programmed differentiation after antigenic stimulation[1,29–32]. Although various intrinsic and extrinsic cues integrated during the antigen-specific priming and expansion phases variably affect the differentiation of responding T cells and ultimately induce heterogeneous effector populations[4,6,33–38], the exact nature and mechanism of shaping such heterogeneity remain unclear. In this study, we demonstrate atypical IL-17-producing effector differentiation for CD8$^+$ T cell compartment and emphasized the previously unappreciated heterogeneity that is inherent to a steady-state pool of naive CD8$^+$ T cells for Tc17 differentiation.

A notable finding of this study is that naive CD8$^+$ T cells are composed of functionally distinct subpopulations with different potential to undergo Tc17 fate, and that these properties are highly associated with the relative self-reactivity of individual naive subsets. Based on the level of CD5 expression, reflecting developmentally acquired intrinsic affinity of TCR for self-ligands, we were able to classify naive CD8$^+$ T cell pools to define CD5$^{lo}$ cells as high Tc17 producers and CD5$^{hi}$ cells (especially Ly6C$^+$ cells) as poor Tc17 producers, which is a crucial mechanistic basis for severe inflammatory diseases in the IBD and GVHD settings. Hence, these findings illustrate a striking connection between the intrinsic self-reactivity of T cells and the skewing of their fate, and provide insights into the additional complexity of naive T cell pools in shaping the functional heterogeneity of effector populations in a diverse range of immune environments.

How does the developmental self-reactivity of naive CD8$^+$ T cells influence their Tc17 potential? As developing thymic T cells receive proper TCR tuning to different degrees depending on the strength of TCR signals derived from positively selected self-ligands[39–46], it could be speculated that variable levels of cell-intrinsic TCR sensitivity are related to differential Tc17 potential. In this regard, we have previously shown that naive CD8$^+$ T cells differ in their responses to TCR stimulation, and that CD5$^{lo}$ cells have higher TCR-induced Ca$^{2+}$ influx and activation of proximal TCR signaling pathways than do CD5$^{hi}$ cells[10,47]. Therefore, we assumed that relatively lower cell-intrinsic self-reactivity (i.e., less tuned developmentally) would paradoxically provide a selective advantage to CD5$^{lo}$ cells over CD5$^{hi}$ cells for Tc17 differentiation. In fact, compared with CD5$^{hi}$ cells, CD5$^{lo}$ cells induce much higher levels of IRF4, a TCR-dependent transcription factor crucial for the initiation of the Tc17 program[48,49], and activation of ERK, a known positive regulator along with the IL-6-STAT3-RORγT axis for Tc17 differentiation[50,51]. Furthermore, we found that such functional distinction between CD5$^{lo}$ and CD5$^{hi}$ cells for Tc17 differentiation was not only limited to polyclonal naive CD8$^+$ T cell populations, but also applied to monoclonal populations and even to human naive T cell populations. Hence, our findings strongly support the notion that different subsets of naive CD8$^+$ T cells have considerably different potential for Tc17 fate, depending on intrinsic self-reactivity, and

suggest that this phenomenon may be an important physiological mechanism.

Unlike CD5$^{lo}$ cells, CD5$^{hi}$ cells with higher TCR self-reactivity could not efficiently differentiate into Tc17 cells. This property, inherent to CD5$^{hi}$ cells, is of particular interest, because this subset has a greater sensitivity to cytokines[7,9], including the Tc17-polarizing cytokines, IL-6 and TGF-β (as assessed by enhanced amounts of p-STAT3 and p-SMAD2/3, respectively). Thus, the weaker TCR response of CD5$^{hi}$ cells (indicated by lower IRF4 expression and ERK activation) appeared to be a key attribute of poor Tc17 differentiation potential, even under conditions of enforced signals driven by Tc17-promoting cytokines. Alternatively, although endogenously produced IL-2 could antagonize Tc17 differentiation predominantly in CD5$^{hi}$ cells compared with that in CD5$^{lo}$ cells, the CD5$^{hi}$ cells had consistently poor Tc17 potential, even in the presence of IL-2 blockade. This further corroborates the importance of the relative intensity of TCR signaling over the benefits of enhanced cytokine signaling.

In addition to the effect of differential TCR sensitivity, we demonstrated a crucial role of SMAD3 in determining the Tc17 differentiation potential of naive CD8$^+$ T cells. Although SMAD3 negatively regulates Th17 cell differentiation[27,52], its physiological role in controlling the functional heterogeneity of naive CD8$^+$ T cells has not yet been deciphered. In this regard, a notable finding in our study was that naive CD8$^+$ T cell populations in a steady state exhibit variable basal expression of SMAD3 (both unphosphorylated and phosphorylated), possibly affecting cell fate decisions, with higher levels in CD5$^{hi}$ cells than in CD5$^{lo}$ cells. The striking outcome of the differential SMAD3 expression is that it is apparently directly responsible for the distinctly higher or lower Tc17 potential in CD5$^{lo}$ and CD5$^{hi}$ cells, with evidence from our retroviral transduction experiments, in which SMAD3 was either overexpressed or knocked down, resulting in functionally decreased and increased Tc17 differentiation, respectively. Furthermore, the observed functional distinction for Tc17 fate between SMAD3$^{lo}$ CD5$^{lo}$ and SMAD3$^{hi}$ CD5$^{hi}$ cells was broadly applicable to the corresponding subsets from both murine (both polyclonal and monoclonal) and human naive T cells. Therefore, allocating variable levels of SMAD3 in a manner dependent on the relative intensity of developmental self-reactivity may serve as a conserved mechanism for controlling the functional heterogeneity of naive CD8$^+$ T cell populations to generate a broader spectrum of Tc17 differentiation potential.

As CD5$^{hi}$ cells have higher ability to respond to homeostatic cytokine cues, the enhanced responses to these cytokines may be responsible for the higher SMAD3 expression in this subset. Of the various cytokines available at steady state, type I IFN is crucial for the formation of the Ly6C$^+$ subset of CD5$^{hi}$ cells[5]. In addition, given our finding that basal levels of phosphorylated SMAD3 were more pronounced in CD5$^{hi}$ (especially Ly6C$^+$) cells than in CD5$^{lo}$ cells, the role of tonic TGF-β signals may also be considered. In fact, it has recently been shown that naive CD8$^+$ T cells are exposed to the active pool of TGF-β provided by integrin αVβ8-expressing migratory dendritic cells in

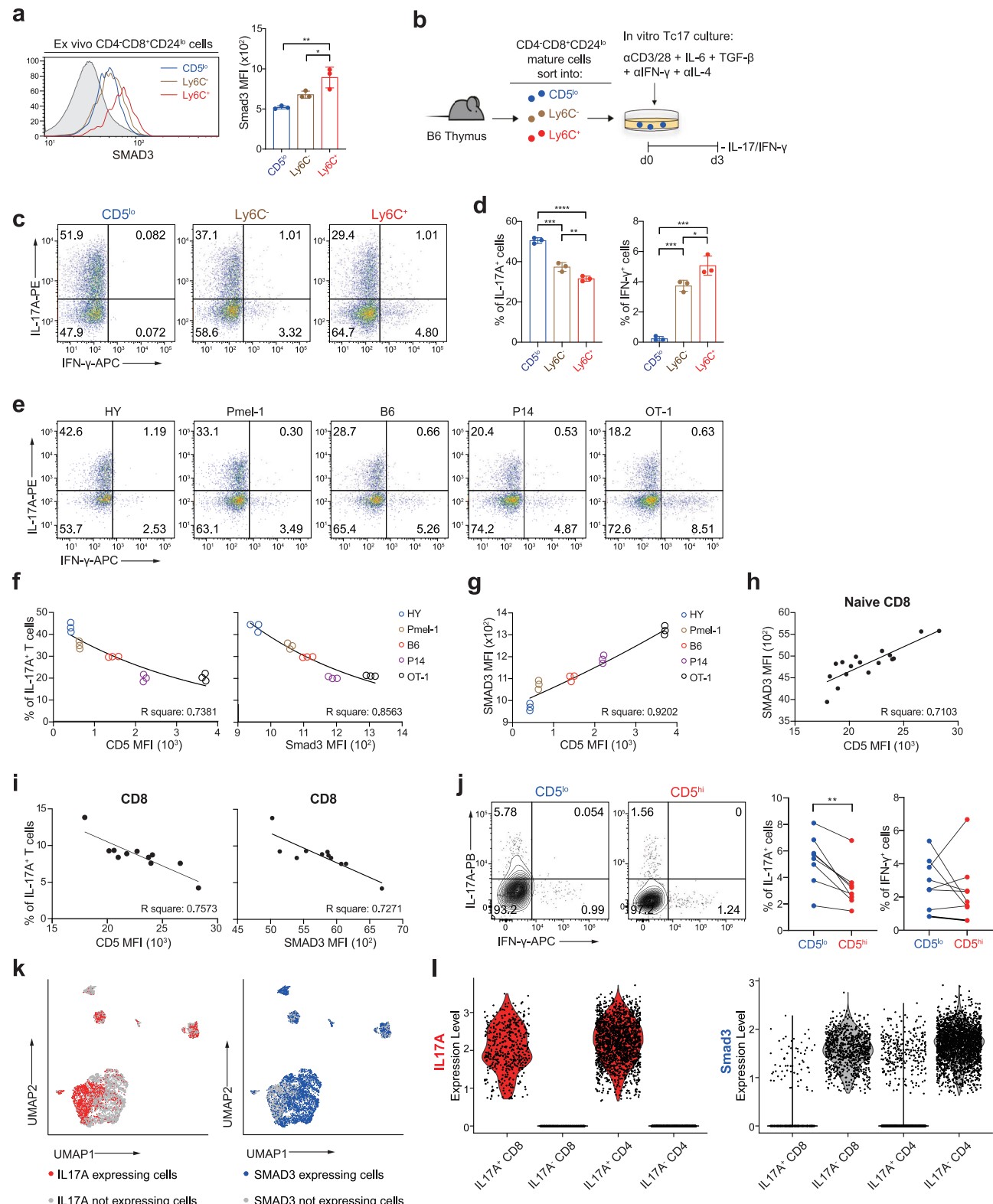

lymph nodes, and that this phenomenon is important for the formation of epithelial tissue-resident memory T (T$_{RM}$) cells upon antigenic stimulation[53]. Therefore, it will be notable to elucidate whether CD5$^{hi}$ (particularly Ly6C$^+$) cells have a higher T$_{RM}$ differentiation potential via more pronounced TGF-β preconditioning. Alternatively, based on the observation that the difference in the expression of SMAD3 was apparent even at the stages of CD8$^+$ SP thymocytes, a cytokine, such as IL-7, which has a wide range of activity during both thymic

development and peripheral homeostasis, may also be a likely candidate, and thus warranting further investigation.

Although CD5$^{lo}$ cells acquire a selective advantage over CD5$^{hi}$ cells to undergo an effector differentiation program in favor of Tc17 fate, the exact physiological significance of why CD5$^{lo}$ cells evolved to better differentiate into Tc17 cells remains unclear. One possible explanation is that the enhanced Tc17 potential of CD5$^{lo}$ cells is perhaps context-dependent and detected only under conditions promoting type 17 but

**Fig. 7 | Developmental self-reactivity is positively correlated with endogenous SMAD3 levels and is negatively associated with Tc17-skewing property.**
**a** Endogenous SMAD3 levels in CD5$^{lo}$, Ly6C$^-$, and Ly6C$^+$ subsets from B6 mature CD4$^-$CD8$^+$CD24$^{lo}$ thymocytes. **b** Experimental scheme for **c, d**.
**c, d**, Representative FACS plots for IL-17A/IFN-γ production (**c**) and the percentage of IL-17A$^+$ and IFN-γ$^+$ cells (**d**) after Tc17-polarizing culture with thymic CD8$^+$ T$_N$ subsets. **e–g** Representative FACS plots for IL-17A/IFN-γ production (**e**), relationship between CD5 (left) or SMAD3 (right) levels ex vivo and the percentage of IL-17A$^+$ cells after Tc17-polarizing culture with HY, Pmel-1, B6, P14, and OT-1 CD8$^+$ T$_N$ populations (**f**), and relationship between CD5 and SMAD3 levels ex vivo (**g**). **h** Relationship between CD5 and SMAD3 levels in CD8$^+$ T$_N$ cells (gated on CD3$^+$CD8$^+$CCR7$^+$CD45RA$^+$) from healthy human bloods ($n = 17$). **i** Relationship between CD5 (left) or SMAD3 (right) levels ex vivo and the percentages of IL-17A$^+$ cells after Tc17-polarizing cultures with human CD8$^+$ T$_N$ populations ($n = 11$).

**j** Representative FACS plots for IL-17A/IFN-γ production (left) and the percentage of IL-17A$^+$ and IFN-γ$^+$ cells (right) after Tc17-polarizing culture with either human (CD3$^+$CCR7$^+$CD45RA$^+$CD95$^-$) CD5$^{lo}$ or CD5$^{hi}$ CD8$^+$ T$_N$ cells (left) ($n = 8$). Data are pooled from two independent experiments ($n = 3$/experiment). **k** *IL17A*- (left) and *SMAD3*-expressing (right) cells in *CD3$^+$* cells from public scRNA-seq dataset (GSE162335) of UC patients' tissues were highlighted in the UMAP. **l,** *IL17A* (left) and *SMAD3* expressions (right) of *IL17A$^+$CD8$^+$*, *IL17A$^-$CD8$^+$*, *IL17A$^+$CD4$^+$*, and *IL17A$^-$CD4$^+$* cells. Data are representative of three independent experiments (**a, d, f, g**, $n = 3$ per experiment) or pooled from two independent experiments and presented as the mean ± SD (**a, d**) and ± SEM (**j**). Statistical significance by two-way ANOVA Multiple comparisons (**a, d**), simple linear regression (**f–i**), and two-tailed paired *t*-test (**j**). $^*P < 0.05$; $^{**}P < 0.01$; $^{***}P < 0.001$; $^{****}P < 0.0001$. Source data and exact *P* value are provided as a Source Data file.

not type 1 immunity (for example, CD5$^{lo}$ cells transferred into *Rag-1$^{-/-}$* hosts in a viral infectious setting). Therefore, we propose that CD5$^{hi}$ cells, presumably due to an enhanced response to inflammatory cytokines[5,9,10,54], are better poised to differentiate into Tc1 cells, exhibiting greater expansion and potent cytotoxic activity to eliminate intracellular pathogens[5,6,8,55]. In contrast, CD5$^{lo}$ cells are poised to differentiate into Tc17 cells with strong immunomodulatory activity, specialized in clearing extracellular pathogens, and, if not properly controlled, cause immunopathology[56]. It is thus inferred that the immune system may develop a mechanism to separate functional labor so that the right subsets of naive CD8$^+$ T cells can be deployed, providing a reasonable explanation for why both CD5$^{lo}$ and CD5$^{hi}$ cells need to be physiologically selected during T cell development.

Hence, these findings further extend the conceptual framework that the thymic positive selection process is required to generate a functionally heterogeneous repertoire of naive T cells beyond a given TCR specificity. In the future, it will also be interesting to address whether the highly poised Tc17 potential of CD5$^{lo}$ cells and its association with severe inflammatory diseases observed in IBD and GVHD models extend to other disease settings, such as autoimmunity and cancer. This information, together with similar experimental approaches for other effector types, such as Tc2 and Tc9, as well as for subsets of naive CD4$^+$ T cells, will provide valuable insights for better understanding the complex immune and/or pathophysiological responses to various acute and chronic inflammatory diseases.

## Methods
### Mice
C57BL/6 (B6), and BALB/c mice were purchased from The Jackson Laboratory. B6.SJL (Ly5.1), B6.PL (Thy1.1), P14, OT-I, HY, and *Rag1$^{-/-}$* mice (all on a B6 background) were kindly provided by Dr. Kwang Soon Kim (Pohang University of Science and Technology, Pohang, Korea), and *Ccr6$^{-/-}$* mice were kindly provided by Dr. You-Me Kim (Korea Advanced Institute of Science and Technology, Daejeon, Korea). All mice were maintained under specific pathogen-free conditions at the Laboratory Animal Center of Chonnam National University (CNU) Medical School. The mouse housing conditions were maintained at 21–23 °C room temperature (RT), 50–70% relative humidity, and 12 h light/dark cycle. Both male and female mice, unless described otherwise, at 6 to 8 weeks of age were used in the experiments. Sex was not considered a biological variable, as no differences related to the overall immune contexts were observed between the male and female mice. All animal experiments were performed in accordance with protocols approved by the Animal Experimental and Ethics Committee at CNU Medical School (ethics approval no. CNU IACUC-H-2022-16).

### Induction of experimental colitis and histological analysis
FACS-sorted CD44$^{lo}$CD5$^{lo}$Ly6C$^-$, CD44$^{lo}$CD5$^{hi}$Ly6C$^-$, and CD44$^{lo}$CD5$^{hi}$Ly6C$^+$ CD8$^+$ T$_N$ subsets from B6. Thy1.1 or B6. Ly5.1 mice were intravenously transferred into *Rag1$^{-/-}$* mice ($2 \times 10^5$ cells/mouse).

Daily body weight changes were measured. Mice were sacrificed on indicated date in this paper. Cellular analysis and histological analysis were proceeded from mLN and large intestine (LI). 5 mm of colon tissues collected from mice were treated with 4% paraformaldehyde (Tech & Innovation) for 24 h and embedded into paraffin block for hematoxylin and eosin (H & E) staining. 5 μm sections of the colon tissue were stained with H&E for histological analysis. Clinical scoring of colitis was graded on a scale of 0–4, average scores of two different parameters, as follows: Epithelial damage score (0: none, 1: minimal loss of goblet cells, 2: extensive loss of goblet cells, 3: minimal loss of crypts and extensive loss of goblet cells, 4: extensive loss of crypts); Infiltration score (0: none, 1: infiltrate around crypt bases, 2: infiltrate in muscularis mucosa, 3: extensive infiltrate in muscularis mucosa and edema, 4: infiltration of submucosa). Histological evaluation was conducted in a blinded fashion. For immunofluorescence imaging, freshly harvested colon tissue was fixed in 4% PFA for 1 h and then embedded into O.C.T. Compound (Sakura). 5 μm sections of the fresh frozen tissue were blocked with 10% normal mouse serum in PBST (containing 0.3% Tripton × 100) for 1 h at RT. After incubation with fluorophore-labeled primary antibodies for 1 h at RT, nuclei were stained with DAPI (ThermoFisher), and coverslips were mounted with ProLong™ Gold Antifade Mountant (ThermoFisher). Images were taken on TCS SP5 – Leica Application Suite X version 3.7.4.23463 (Leica, Germany).

### Serum ELISA
For detection of in vivo cytokines, sera from the indicated mice were collected and analyzed by a standard protocol using a cytokine sandwich ELISA kit according to the manufacturer's instructions. The following ELISA kit (R&D Systems) were used: IL-1β (MLB00C), IL-6 (M6000B), and TNF-α (MTA00B).

### Induction of GVHD
BALB/c mice were received 1,000 cGy total body X-ray irradiation (X-RAD 320; PRECISION). T cell-depleted wild-type B6 bone marrow cells alone or with FACS-sorted CD44$^{lo}$CD5$^{lo}$Ly6C$^-$, CD44$^{lo}$CD5$^{hi}$Ly6C$^-$, and CD44$^{lo}$CD5$^{hi}$Ly6C$^+$ CD8$^+$ T$_N$ subsets from B6.Thy1.1 mice were adoptively transferred into 24 h after lethally irradiated recipient mice. Daily body weight changes and survival rate were followed. GVHD-induced mice were sacrificed at day 7 for cellular analysis, and at day 15 for the histological analysis of kidney and LI. Liver and kidney collected from mice were treated with 4% paraformaldehyde (Tech & Innovation) for 24 h and embedded into paraffin block for hematoxylin and eosin (H & E) staining. 5-μm sections of tissue were stained with H&E for histological analysis.

### Flow cytometry
Cell suspensions were washed with FACS staining buffer (PBS containing 2% FBS and 0.05% sodium azide). Cells were stained for 15 min on ice with propidium iodide (Invitrogen) to distinguish live and dead

cells and with anti-CD16/32 Fc receptor block (Invitrogen) to prevent non-specific antibody binding. Cells were then washed twice and stained with fluorochrome-conjugated antibodies for 20 min on ice at the specified dilutions. For intracellular cytokine stain, freshly harvested in vitro polarized cells or isolated cells from various tissues were cultured with cell stimulation cocktail plus protein transport inhibitors (Invitrogen) for 4 h. Cultured cells were washed twice and the surface staining was performed as described above. Then cells were fixed and permeabilized with the Cytoperm/Cytofix kit (BD Pharmingen) for 20 min on ice. Cells were then washed twice with BD Perm/Wash™ Buffer and stained with fluorochrome-conjugated antibodies for 30 min on ice at the specified dilutions. For intracellular transcription factor stain, freshly harvested in vitro polarized cells or isolated cells from various tissues were washed twice and the surface staining was performed as described above. Cells were fixed and permeabilized with the fixation/permeabilization reagent (eBioscience) for 30 min on ice. Cells were then washed twice with permeabilization buffer and stained with fluorochrome-conjugated antibodies for 30 min on ice. For stains that used fluorochrome-unconjugated primary antibodies, cells were fixed using BD Cytofix™ for 30 min and permeabilized using chilled 90% methanol for 1 h. Fixed and permeabilized cells were stained with primary antibodies and then stained with fluorochrome-conjugated secondary antibody (goat anti-Rabbit IgG). The antibodies used for these studies are listed in Supplemental Table 1. Flow cytometry samples were run using a LSRII or FACS Canto II (BD Biosciences) and analyzed by FlowJo v10.6.2 software (Tree Star).

## Cell sorting

The surface staining for isolated single cells was performed with fluorochrome-conjugated anti-CD8 (clone 53-6.7), anti-CD44 (clone IM7), anti-CD62L (clone MEL-14), anti-CD5 (clone 53-7.3), and anti-Ly6C (clone HK1.4). Stained cells were sorted as indicated using a BD FACSAria III or Beckman Coulter CytoFLEX SRT cell sorter. CD8$^+$ cells were gated first, and then gated as CD44$^{lo}$CD62L$^{hi}$ population to define naive T cells. These cells were further gated based on CD5 and Ly6C expression to separate into CD5$^{lo}$, CD5$^{hi}$Ly6C$^-$ and CD5$^{hi}$Ly6C$^+$ cells with an average of > 99% purity.

## In vivo T-cell proliferation

FACS-sorted CD44$^{lo}$CD5$^{lo}$Ly6C$^-$, CD44$^{lo}$CD5$^{hi}$Ly6C$^-$, and CD44$^{lo}$CD5$^{hi}$Ly6C$^+$ CD8$^+$ T$_N$ cells were labeled with 2.5 μM of Cell Trace™ Violet (CTV; ThermoFisher). Each CTV-labeled subset was adoptively transferred into an individual sex and age matched Rag1$^{-/-}$ recipient mice. Cells were isolated from mLN 7 days after transfer and analyzed by flow cytometry. For BrdU incorporation assay, mice were injected intraperitoneally (i.p.) with BrdU (1 mg/mouse) and then administered with drinking water containing BrdU (0.8 mg/ml) for 2–3 days before sacrifice. BrdU staining was performed with eBioscience™ BrdU Staining Kit for Flow Cytometry FITC (eBioscience) according to manufacturer's protocol. In brief, cells isolated from LI were stained with surface markers, then fixed and treated with DNase I. Cells were stained with fluorochrome-conjugated anti-BrdU antibody and analyzed with flow cytometry.

## Gut permeability assay

In vivo gut permeability assay for mice adoptively transferred with individual CD8$^+$ T$_N$ subsets was conducted at the indicated dates after 4 h of oral administration of FITC-dextran (Merck). The serum FITC concentration was measured using blood collected by retro-orbital bleeding, and fluorescence was measured using a spectrofluorometer.

## In vitro transwell migration assay

Chemotactic activity was tested using 5 μm-pore polycarbonate membrane inserted into transwell cell culture chambers (Corning, USA). FACS-sorted CD44$^{lo}$CD5$^{lo}$Ly6C$^-$ and CD44$^{lo}$CD5$^{hi}$Ly6C$^+$ CD8$^+$ T$_N$

cells from B6. Thy1.1 were transferred into Rag1$^{-/-}$ mice. After 7 days, donor cells in mLN were stained with anti-Thy1.1 and labeled with or without CTV. CTV-labeled and unlabeled subsets mixed in 1:1 ratio at a density of $1 \times 10^6$ cells per well in 48-well plates. $1 \times 10^6$ cells per 200 μl of aliquots of cell suspension were added to the upper compartment, while the lower compartments contained 600 μl of medium with 10 ng/ml CCL4 or 10 ng/ml CCL20. After incubation for 4 h at 37 °C, frequency of migrated cells was calculated as follow: bottom chamber cell counts / (bottom chamber cell counts + upper chamber cell counts) × 100.

## In vitro Tc17 cell polarization of murine CD8$^+$ T$_N$ cells

FACS-sorted CD44$^{lo}$CD5$^{lo}$Ly6C$^-$, CD44$^{lo}$CD5$^{hi}$Ly6C$^-$, and CD44$^{lo}$CD5$^{hi}$Ly6C$^+$ CD8$^+$ T$_N$ cells were cultured in anti-CD3ε (2.5 μg/ml) (clone 145-2C11) and anti-CD28 (1 μg/ml) (clone 37.51) coated Nunc MaxiSorp™ Flat-Bottom Plate with recombinant mIL-6 (20 ng/ml) (PeproTech), rmTGF-β (2.5 ng/ml) (PeproTech), anti-IFN-γ (10 μg/ml) (clone XMG1.2), and anti-IL-4 (10 μg/ml) (clone 11B11) for 3 days.

## Purification of human naive T cells

Healthy human blood samples were provided by the Korean Red Cross in accordance with a protocol approved by the Institutional Review Boards of Chonnam National University Hwasun Hospital (CNUHH-2021-045). The blood used in this study was granted a written informed consent exemption by the Institutional Review Board in accordance with the Korean Bioethics and Safety Act on the ground that it was not feasible to obtain written consent from the research subjects, there was no reason to presume refusal of consent by the research subjects, and the risk to the research subjects was extremely low. The information about the blood donors were inaccessible due to local regulations, therefore, sex, gender, and age were not tracked as biological variables. Peripheral blood mononuclear cells (PBMC) were purified using Lymphoprep (Alere Technologies) and were FACS-sorted into either CCR7$^+$CD45RA$^+$CD95$^-$CD4$^+$ or CD8$^+$ T$_N$ cells. In some experiments, CCR7$^+$CD45RA$^+$CD95$^-$CD8$^+$ T$_N$ cells were further sorted into CD5$^{lo}$ and CD5$^{hi}$ subsets based on the lower or upper 20% of CD5 expression using a BD FACSAria III or Beckman Coulter CytoFLEX SRT cell sorter.

## In vitro Th17/Tc17 cell polarization of human naive T cells

FACS-sorted CD4$^+$ or CD8$^+$ (and CD5$^{lo}$ or CD5$^{hi}$) T$_N$ cells ($-1-2 \times 10^4$/well) were cultured with Dynabeads™ (Gibco) in 24-well plate (Effendorf) with recombinant hIL-1β (50 ng/ml) (PeproTech), hTGF-β (2 ng/ml) (PeproTech), hIL-23 (50 ng/ml) (PeproTech), anti-hIFN-γ (10 μg/ml) (clone B27), and anti-hIL-4 (10 μg/ml) (clone MP4-25D2) for 10 days. Cells were replaced with fresh culture medium every 2–3 days while in culture, and at each replacement, cells were collected, counted, and re-cultured $2 \times 10^6$/well under the same conditions until intracellular cytokine assays were performed.

## Western blot

Cells cultured under the conditions indicated were harvested, washed with ice-cold PBS, and lysed on ice for 15–30 min in a lysis buffer (20 mM Tris, pH 7.5, 150 mM NaCl, 1 mM EDTA, 1 mM EGTA, 1% Triton X-100, 2.5 mM sodium pyrophosphate, 1 mM β-glycerophosphate, 1 mM Na$_3$VO$_4$, 1 mM PMSF, and 1 μg/ml aprotinin and leupeptin). Cell lysates were resolved by 4–12% Bis-Tris SDS-PAGE Gel (Invitrogen), transferred onto nitrocellulose membrane (Invitrogen), blocked with 5% dry non-fat milk in Tris buffered saline (pH 7.4) containing 0.1% Tween-20, and probed with the following Abs to (Abs were used at 1:1000 and purchased from Cell Signaling Technology unless otherwise described): p-STAT3 (Tyr705; Cell Signaling Technology), p-SMAD2 (Ser465/467; Cell Signaling Technology), p-SMAD3 (Ser423/425; Cell Signaling Technology), and β-actin (Sigma-Aldrich; 1:10,000). Immunoreactivity was detected by SuperSignal™ West Pico

Chemiluminescent Substrate (Thermo Scientific) and SuperSignal™ West Femto Chemiluminescent Substrate (Thermo Scientific) with LAS-3000 (FujiFilm).

## RNA isolation, cDNA synthesis, and qRT-PCR
CD44$^{lo}$CD5$^{lo}$Ly6C$^-$, CD44$^{lo}$CD5$^{hi}$Ly6C$^-$, and CD44$^{lo}$CD5$^{hi}$Ly6C$^+$ CD8$^+$ T$_N$ cells were sorted from pulled spleens and LNs. RNA was isolated with NuleoZOL (Macherey-Nagel) and extracted RNA was measured by NanoDrop and 1 µg of RNA was reverse transcribed into cDNA with M-MLV reverse transcriptase and oligo dT (TAKARA), following manufacturer's instructions. Real-time RT-PCR was performed with the TaqMan Gene Expression Master Mix using StepOnePlus Real-Time PCR System version 2.2.2 (Applied Biosystems). The following TaqMan probes (Applied Biosystems) were used: *Rorc* (Mm01261022), *Il17a* (Mm00439618), *Irf4* (Mm0516431), *Ifnγ* (Mm01168134), *Tbx21* (Mm0045960), *Eomes* (Mm01351984), and *Rn18sRn45s* (Mm03928990). The expression level was calculated as 2-DCT, where DCT is the difference between the CT values of the target gene and 18S.

## Bulk RNA-seq, public scRNA-seq, and bioinformatics
FACS-sorted CD44$^{lo}$CD5$^{lo}$Ly6C$^-$, CD44$^{lo}$CD5$^{hi}$Ly6C$^-$, and CD44$^{lo}$CD5$^{hi}$Ly6C$^+$ CD8$^+$ T$_N$ cells were polarized into Tc17 for 3 days. For bulk RNA-seq, at least $1 \times 10^6$ purified cells were used for RNA extraction with a NucleoZOL (Macherey-Nagel). Library preparation was carried out with TruSeq Stranded mRNA Sample Preparation Kit (RS-122-2101-2, Illumina) according to the manufacturer's protocol. RNA-seq was carried out with NextSeq 500 Sequencing System (SY-415-1001, Illumina) using NextSeq 500 HighOutput kit (150 cycles) Reagent Cartridge (15057931, Illumina). Gene Set Enrichment Analysis (GSEA) was performed using GSEA tool version 4.0.1 (Broad Institute). The following options were selected for analysis: Number of permutations, 1000; Collapse dataset to gene symbols, false; Permutation type, phenotype; Enrichment statistic, weighted; Metric for ranking genes, Diff_Of_Classes; Max size, 500; and Min size, 5. Genes with fold changes between samples greater than 2 are selected as differentially expressed genes (DEGs). Expression values of DEGs were converted to z-score and used for generating Heatmap by Prism (GraphPad Software). In some analysis performed on human samples, publicly available scRNA-seq dataset (GSE162335, GSE163314, GSE151177, GSE116222) were analyzed using Seurat in R version 4.3.0. In brief, the droplet-based single-cell RNA-sequencing method produces zero-inflated data, where most transcriptional expressions are zero, while actively expressed genes have values greater than zero. The expression data undergo preprocessing steps called "normalization" and "scaling". After these steps, we analyzed the expression of (A) *CD3E*, (B) *IL17A*, and (C) *SMAD3*. The majority of cells exhibited zero expression, while some cells showed meaningful expressions. The threshold of 0.1 was chosen to distinguish between "negative" and "positive" populations, corresponding to "zero expression" and "over-zero expression", respectively. After normalization and scaling, *IL17A* > 0.1 or *SMAD3* > 0.1 cells from *CD3E* > 0.1 were selected for further analysis. *IL17A* and *SMAD3* expressions were compared between *IL17A* positive (*IL17A* > 0.1) and *IL17A* negative (*IL17A* < 0.1) cells. For some analysis, *CD8A* > 0.1 and *CD4* > 0.1 cells were considered CD8 T cells and CD4 T cells, respectively. The detailed information for the public scRNA-seq dataset is as follows:

GSE162335: Transcriptome profiles of CD45$^+$ cells from the lamina propria of 5 pouches without inflammation, 10 pouches with pouchitis, and 11 colons with ulcerative colitis.

GSE163314: Transcriptome profiles of CD45$^+$ cells from colon biopsy tissue of 2 Crohn's disease patients, 2 Spondyloarthritis patients, and 2 Crohn's disease/Spondyloarthritis co-morbid patients.

GSE151177: Transcriptome profiles of inflammatory cells emigrating from a punch biopsy of 13 human Psoriasis lesional skin and the normal control skin of 5 healthy volunteers.

GSE116222: Transcriptome profiles of cells isolated from colonic biopsies collected from 3 healthy individuals and 3 patients with ulcerative colitis from an inflamed area of colon and adjacent non-inflamed area.

## ChIP assay
ChIP assay was done according to the manufacturer's instructions (Cell Signaling Technology). In brief, CD44$^{lo}$CD5$^{lo}$Ly6C$^-$, CD44$^{lo}$CD5$^{hi}$Ly6C$^-$, and CD44$^{lo}$CD5$^{hi}$Ly6C$^+$ CD8$^+$ T$_N$ cells were sorted from B6 mice and cultured under Tc17 polarizing condition for 1 and 3 days. Cells were fixed with 16% of paraformaldehyde (Electron Microscopy Sciences), resuspended in the supplied ChIP buffer and lysed by sonication. Antibodies for IRF4 (D9P5H; Cell Signaling Technology), or SMAD3 (C67H9; Cell Signaling Technology) were added to the ChIP sample followed by incubation of the samples overnight at 4 °C with rotation. ChIP-grade Protein G magnetic beads (Cell Signaling Technology) were added to the samples and the tubes were placed in a Magnetic Separation Rack (Invitrogen) to isolate the antibody-bound chromatins. Chromatins were eluted from the antibody/magnetic beads by elution buffer at 65 °C with vortexing and purified in the spin column. Real-time PCR was done with purified DNA and previously reported primers[27,48] for the regulatory loci or promotor of *Rorc* and *Il17a* genes. Primer sequences used as a negative control are described in Supplemental Table 2.

## Retroviral vectors for SMAD3 over-expression and short-hairpin (sh) RNA knock-down
For over-expression, full-length *SMAD3* cDNA from mouse lymphocyte was cloned into the EcoRI and XhoI sites of the MigR1 plasmid (Addgene; #27490). shRNA targeting *SMAD3* was cloned in to retroviral shRNA LMP plasmid (Addgene; #36955). Plasmids were transfected to the Platinum E cell line with Fugene HD (Promega) and supernatant containing viral particles were collected after 48 h. Retro-X™ Concentrator (Takara) was added 3 volumes to harvested supernatant and incubated with rotation for 1 day at 4 °C. Sorted CD44$^{lo}$CD5$^{lo}$Ly6C$^-$, CD44$^{lo}$CD5$^{hi}$Ly6C$^-$, and CD44$^{lo}$CD5$^{hi}$Ly6C$^+$ CD8$^+$ T$_N$ cells were cultured in plate-bound anti-CD3ε and anti-CD28 for 16 h and transduced with retroviral supernatants with spin infection with polybrene (8 µg/ml; Merck). Cells were then washed, and further cultured under Tc17 polarizing condition for additional 3 and 5 days for over-expression and shRNA knock-down, respectively. Designed primer sequences are described in Supplemental Table 2.

## Statistical analysis
Statistical significance was calculated using the Prism 8.4.1 software (GraphPad). The nonparametric two-tailed Mann-Whitney test or Tukey's test was used to determine the significance for body weight; all other data was analyzed using two-tailed unpaired Student's *t* test or two-way analysis of variance (ANOVA) as appropriate. All *P* values of ≤0.05 were considered significant.

## Reporting summary
Further information on research design is available in the Nature Portfolio Reporting Summary linked to this article.

# Data availability
Bulk RNA-Seq data has been deposited in the NCBI's Gene Expression Omnibus (GEO) database under the primary accession number GSE240440. Following public scRNA-seq datasets can be accessed with the following links provided: GSE162335, GSE163314, GSE151177, and GSE116222. The authors declare that all other data supporting the findings of this study are within the article and its Supplementary Information file or available from the corresponding author upon request. A Figshare archive is present with this paper https://doi.org/

10.6084/m9.figshare.25053977. Source data are provided with this paper.

## Code availability
We did not generate any new code in this study.

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

## Acknowledgements

We thank Dr. K.S. Kim (Pohang University of Science and Technology) for providing several strains of mice indicated; M.J. Ryu and S.M. An (CNU) for administrative assistance; CNU flow cytometric core facilities for assistance with cell sorting; and H.W. Ryu and M.S. Kim (CNU) for mice breeding and care. We also thank the Korean Red Cross for providing biospecimens used for this study. This work was supported by a grant from the National Research Foundation (NRF) funded by the Korean Ministry of Science and ICT (2020R1A5A2031185, 2020M3A9G3080281 and 2022R1A2C2009385 to J.H.C.) and a grant (HCRI 19001-1*HCRI20012 to D.H.Y and J.H.C.) of CNU Hwasun Hospital.

## Author contributions

G.W.L. and J.H.C. initiated and designed the main idea of this study. G.W.L. performed all major experiments; Y.J.K. and S.W.L. performed experiments using human blood samples; D.K. and H.O.K. performed tissue collection and preparation; J.Y.K. and Y.M.K. supported experiments using *Ccr6*<sup>−/−</sup> mice; S.W.L. and K.K. performed bioinformatics analysis; J.H.R., I.J.C., W.K.B., I.J.O., and D.H.Y. supported interpretation of clinical data and provided critical comments; and G.W.L. and J.H.C. analyzed and interpreted data and wrote the manuscript.

## Competing interests

The authors declare no competing interests.
