## [Peer Review File · Nature Communications]

Developmental self-reactivity determines pathogenic Tc17 differentiation potential of naive CD8+ T cells murine models of inflammationREVIEWER COMMENTS

Reviewer #1 (Remarks to the Author):

The author has done a great deal of work to reveal the role of heterogeneity of naive CD8+ T cells in determining pathogenic type 17 cytotoxic (Tc 17) in the context of inflammatory disease models. These findings provide insight into the mechanism of how heterogeneous T cell immunity can be shaped in a steady-state condition and a better understanding of immune responses to various inflammatory diseases. The authors please address the following concern.

1. The definition of the naïve CD8 T cells. CD44-CD62L+ were considered the markers of naïve T cells mostly. The expression of CD62L should be checked in their naïve CD8 T cells. Or they should gate CD44-CD62L+ first, then further gate CD5 low, CD5high Ly6C- and CD5high Ly6C+. It needs to be seriously considered the naïve cells are really naïve, not the contamination of other population, such as stem cell-like memory cells.
2. The proliferation assay in vivo has demonstrated the difference among the three groups (extended data fig2a). The % of so call Fast cells, which means the proliferated cells upon antigen stimulation, showed a significant difference. Even for the “Slow” part, the MFI of each peak was shift, which means the proliferation upon the hemostasis stimulation was also not the same. So, the conclusion of “... their cell division profiles were also comparable” was not solid. Actually, the activation and proliferation issues could be quickly checked in vitro. It will be helpful to check the activation markers and CTV labelled cells in vitro, and checked in series time points.
3. Please note the problems with abbreviation. I) Some abbreviations were defined repeatedly in the text, such as ChIP, MFI, ect . II) The definition of some abbreviations is not written the full name, such as PBMC, MFI, ect. Please check and revise.
4. Whether the relationship of IRF4 obtained in mice also applies to humans?
5. Figure
 - 1) The scale of both HE and IF images are not clear. Fig.1b, 2d, 3b, 3f, and 4d.
 - 2) The results of statistical analysis were not labeled in Fig.4c, 5f, 7f,7g,7h, and 7i.
 - 3) Fig.4b, why is body weight expressed in percentages instead of mass units (gram)? Or is it the change of body weight?
 - 4) Fig.6a, provide the column graph of MFI analysis of Fig.6a left.
 - 5) Fig.6e and 6f, what did the “GFP- “and “GFP+” represent? It’s confusing. If the vector has GFP fluorescent tag, the vector control should also be GFP+. Why choose GFP+ and GFP- for comparison?
 - 6) Please provide the column graph of MFI analysis of Fig.7a.
 - 7) Please add the R square of Fig.7f and 7g.
6. Please provide information on the samples of the gene expression profile datasets GSE162335, GSE163314 and GSE151177.
7. MATERIALS AND METHODS
 - 1) What is the method of IBD model establishment?
 - 2) What are the conditions for cell culture in vitro?
 - 3) Methods and materials for ELISA are missing.
 - 4) Please provide the probe sequence of qRT-PCR in this study.
 - 5) Line 600-602, which parameter is used to determine negative and positive?
 - 6) Line 622-623, please provide the Addgene number of the plasmid used.
 7. Others
 - 1) Line 115, “proportions of IL-17A+IFN- γ - (and IL-17A+IFN- γ +) cells”. Parentheses are redundant.

- 2) Line 501, "transferred. into Rag1-/- mice." should be "transferred into Rag1-/- mice."
- 3) Line 515, "with anti-CD16/32; Fc block". Semicolon is redundant. And it should be Fc receptors (FcR) block rather than Fc block.

Reviewer #2 (Remarks to the Author):

In the manuscript titled "Developmental self-reactivity determines the pathogenic Tc17 differentiation potential of naïve CD8 T cells by adjusting endogenous SMAD3 expression", Lee et al report their identification of a subset of naïve T cells skewed for Tc17 differentiation based on CD5 expression level. The authors report that the different developmental capacity of the three naïve CD8 T cell subsets to undergo Tc17 differentiation in both IBD and GVHD murine models. Specifically, the authors demonstrate that the CD5^{lo} subset produces higher IL-17A which in turn exacerbates IBD and GVHD pathologies. Further, the authors show, using the IBD model, that migration of CD5^{lo} cells occurs through the CCR6 chemokine axis. Analysis of the cell-intrinsic mechanism for naïve T cell Tc17 skewing revealed that TCR strength/IL-2 signaling and SMAD3 expression are coupled to the generation of Tc17 cells in vivo and in vitro. Finally, using human T cells, the authors validated this novel immunological concept. Overall, this body of work robustly outlines the differential fate-skewing of naïve CD8 T cell subsets and the impact this can have on IBD and GVHD pathology in mouse systems. The overall study is designed very well including multiple mouse models of disease, different TCR tg model, as well as validation in human T cells. The mechanistic insights provided in this study are important but could be further strengthened with the inclusion of some additional data and controls described below.

Major points

This study provides valuable insight into the different Tc17 differentiation capacity of CD5^{lo} naïve T cells in a SMAD3 dependent manner, and potential to establish autoimmunity in murine disease models. However, the link to human disease settings needs to be strengthened. The authors report a strong inverse relationship between IL-17A and SMAD3 expression in UC patient's scRNA data (Fig7 k, l), but the entire premise of the paper is that the autoimmune association with IL17 comes from CD5^{lo} naïve cells. This reviewer would like to know if the relationship between CD5 expression and IL17 expression also exists in the scRNA seq data. Also, among the UC patients, only 18% of the CD3 T cells expressed either IL17A or SMAD3. Can the authors show a correlation between disease severity and IL-17A or SMAD3 or CD5 levels?

Minor points

1. In Figure 1, in mLN, three different naïve CD8 T cell subsets show a similar frequency and number with similar CTV levels. However, in the colon, they exhibited a different number of cells. Is this difference due to different proliferative or anti-apoptotic capacities among the cells?
2. Clarity on whether the data came from the lymph node or colon would help the reader.
3. In Figure 3, authors show different migratory properties of the 3 naïve CD8 T cell subsets through a CCR6 dependent manner. How about CCR5 and GPR15? Is there any effect when authors use CCR5 or GPR15 KO CD8 T cell?
4. In Figure 3g,h, it would be helpful to add the CCR6+ control group.

Reply to the comments raised by Reviewers:

Reviewer #1 (Remarks to the Author):

The author has done a great deal of work to reveal the role of heterogeneity of naive CD8⁺ T cells in determining pathogenic type 17 cytotoxic (Tc17) in the context of inflammatory disease models. These findings provide insight into the mechanism of how heterogeneous T cell immunity can be shaped in a steady-state condition and a better understanding of immune responses to various inflammatory diseases. The authors please address the following concern.

1. The definition of the naive CD8⁺ T cells. CD44^{lo}CD62L⁺ were considered the markers of naive T cells mostly. The expression of CD62L should be checked in their naive CD8⁺ T cells. Or they should gate CD44^{lo}CD62L⁺ first, then further gate CD5^{lo} and CD5^{hi} Ly6C⁻ and CD5^{hi} Ly6C⁺. It needs to be seriously considered the naive cells are really naive, not the contamination of other population, such as stem cell-like memory cells.

We appreciate and totally agree with the reviewer's valid concern about the potential contamination of memory-phenotype [MP; mostly CD44^{hi}CD62L^{hi} (Tcm) and to a lesser extent CD44^{hi}CD62L^{lo} (Tem) and nearly undetectable CD44^{lo}CD95⁺ (Tscm)] cells in our study (presented for the reviewer's perusal in **Reviewer Fig. 1**). Since the major focus of our study was to define naive CD8⁺ T cell heterogeneity, all of our data were conducted with rigorous characterization and purification procedures to ensure that the cells in question are truly naive and are not memory contaminants.

Reviewer Fig. 1. Gating strategy for naive CD8⁺ T cell subsets. Lymph nodes of B6 mice were

harvested and prepared into single cells, then stained with fluorochrome conjugated α CD8, α CD44, α CD62L, α CD5, and α Ly6C, and analyzed for defining naive subsets by flow cytometry.

So, as the reviewer pointed out, in all our experiments with FACS-purified naive CD8⁺ T cell subsets (from normal B6 and various gene-modified mice indicated), CD44^{lo}CD62L^{hi} cells were first defined as a naive and gated stringently for cell sorting to clearly separate them from CD44^{hi}CD62L^{lo/hi} cells, and then to further separate into CD5^{lo}, CD5^{hi}Ly6C⁻ and CD5^{hi}Ly6C⁺ cells with an average of > 99% purity. FACS data before and after cell sorting are presented for the reviewer's perusal in **Reviewer Fig. 2**, and also added as **Supplementary Fig. 1a** in the revised manuscript.

Reviewer Fig. 2. Sorting purity of naive CD8⁺ T cells subsets. Lymph nodes of B6 mice were harvested and prepared into single cells, then stained with fluorochrome conjugated α CD8, α CD44, α CD62L, α CD5, and α Ly6C for FACS sorting. Purity for each purified subset gated was regularly checked to be >99%.

To further address the reviewer's concerns about the true naivety of the above three subsets used in this study, we also performed additional *in vitro* experiments and now summarize these new data as follows:

1) Since memory (and MP) cells are known to functionally differ from naive cells with faster and greater activation signatures upon TCR stimulation, we compared these memory features with naive subsets investigated in our study. For this, FACS-purified CD44^{hi} MP CD8⁺ T cells and three naive (CD44^{lo}) CD8⁺ T cell subsets (i.e., CD5^{lo}, CD5^{hi}Ly6C⁻ and CD5^{hi}Ly6C⁺) were stimulated for 5 hours with plate-bound anti-CD3/CD28 mAbs, followed by conducting bulk RNA-seq to analyze early activation gene expression profiles and to compare with publicly available dataset for memory CD8⁺ T cells (GSE10239). The results showed that all three naive CD8⁺ T cell subsets (including Ly6C⁺ cells) closely resembled each other and were distinctly different from CD44^{hi} MP cells (presented for the reviewer's perusal in

Reviewer Fig. 3a, b). Notably, the levels of some early effector genes, such as *Il2*, *Ifng* and *Gzmb*, were still very low in all three naive subsets, compared to those observed in MP cells, analyzed even for this short time period after stimulation (**Reviewer Fig. 3c**). These results were in good agreement with previous reports demonstrating that MP cells exhibit faster and more robust activation than naive cells upon TCR stimulation (Cho et al, PNAS 1999; Veiga-Fernandes et al, Nat Immunol 2000; Kersh et al, J Immunol 2003; DiSpirito et al, Cell Res 2010), and based on these criteria, we believe that CD5^{lo}, CD5^{hi}Ly6C⁻ and CD5^{hi}Ly6C⁺ subsets are functionally naive and clearly different from MP cells.

Reviewer Fig. 3. Gene expression profiles for purified MP and naive CD8⁺ T cell subsets. Naive CD8⁺ T cell subsets (CD5^{lo}, CD5^{hi}Ly6C⁻ and CD5^{hi}Ly6C⁺) and CD44^{hi} MP cells were FACS-purified, then stimulated with plate-bound anti-CD3/CD28 for 5 h. Cells were then used for bulk RNA-seq analysis. **(a,b)** Publicly available dataset for memory CD8⁺ T cells (Genes upregulated in memory CD8⁺ T cells compared to naive CD8⁺ T cells; GSE10239) were used. **(a)** Genes in the GSE10239 gene set were collected from our RNA-seq data and used for heat map. **(b)** Gene set enrichment analysis (GSEA) were carried out with Broad Institute GSEA software using the GSE10239 gene set and our RNA-seq data. **(c)** Various activation and early effector genes including *Ifng*, *Il2*, and *Gzmb*

were analyzed.

2) We also compared proliferative ability of FACS-purified MP cells and three naive CD8⁺ T cell subsets in response to γ c cytokines (IL-2, IL-7 and IL-15), as these cytokines have been shown to induce greater proliferation of MP cells. As expected, MP cells showed markedly enhanced proliferative responses than naive cells when cultured with these cytokines, particularly with IL-2 or IL-15, although there was a moderate response in CD5^{hi} naive subsets by IL-7 (presented for the reviewer's perusal in **Reviewer Fig. 4**). Again, these data support that CD5^{lo}, CD5^{hi}Ly6C⁻ and CD5^{hi}Ly6C⁺ subsets are considered as naive cells that are functionally different from MP cells.

Reviewer Fig. 4. Proliferative responses of purified MP and naive CD8⁺ T cell subsets in response to cytokines. FACS-purified, CTV-labeled naive CD8⁺ T cell subsets (CD5^{lo}, CD5^{hi}Ly6C⁻

and CD5^{hi}Ly6C⁺) and CD44^{hi} MP cells were cultured with cytokines IL-2, IL-7 and IL-15 for 5–7 days. CTV dilutions were analyzed by flow cytometry.

In addition to the above new data, we would also like to mention our previously published observations (Ju et al, Nat Comm 2021) that are relevant to the reviewer's concerns. In agreement with the aforementioned higher proliferation of MP cells upon cytokine stimulation, IFN- γ production on CD8⁺ T cell subsets stimulated for 5 hours with PMA and ionomycin was much greater in MP cells than in three naive subsets for the percentages (~5–38% vs. ~80%) and the mean fluorescence intensity (MFI) (~600–1000 vs. ~4500). These data (Ju et al, Nat Comm 2021) are presented here for the reviewer's perusal in **Reviewer Fig. 5**.

Reviewer Fig. 5. IFN- γ production capacity of MP and naive CD8⁺ T cell subsets. Splenocytes of normal B6 mice were cultured with PMA and ionomycin for 5 hours. **(a)** Proportion of IFN- γ ⁺ cells and **(b)** IFN- γ MFI of IFN- γ ⁺ cells were analyzed for gated MP and naive CD8⁺ T cell subsets by flow cytometry.

With regard to the relative density of cell surface markers (e.g., CD44, CD122 and CD183), we have also shown that the levels of these markers were much lower in three naive CD8⁺ T cell subsets compared to those seen in MP cells (Ju et al, Nat Comm 2021; presented for the reviewer's perusal in **Reviewer Fig. 6**).

Reviewer Fig. 6. Expression of CD44, CD122, CD183, and Ly6C in MP and naive CD8⁺ T cell subsets. Expressions of the (a) CD44, (b) CD122, (c) Ly6C, and (d) CD183 were analyzed in the naïve CD8⁺ T subsets (CD5^{lo}, CD5^{hi} Ly6C⁻, CD5^{hi} Ly6C⁺) and MP from the spleen of WT B6 mice.

Finally, in addition to all these data described above, evidence on the "naive" nature (both for phenotype and function) of the three naive CD8⁺ T cell subsets, especially CD5^{hi}Ly6C⁺ subset, does not simply hinge on the use of polyclonal B6 naive CD8⁺ T cell subsets. In this study, we also provided extensive *in vitro* data with monoclonal naive CD8⁺ T cell subsets from P14 TCR transgenic mice (Fig. 5e and Supplementary Fig. 4c in the revised manuscript). As the P14 mice used in our study were on a *Rag1*^{-/-} background, CD44^{hi} P14 cells were nearly undetectable in these mice (presented for the reviewer's perusal in **Reviewer Fig. 7**). Therefore, we think that a potential contamination of MP cells might be negligible

particularly after our stringent sorting procedure.

Reviewer Fig. 7. Phenotype of CD8⁺ T cells from P14.Rag1^{-/-} mice. Splenocytes of wild-type (WT) P14 and P14.Rag1^{-/-} mice were stained with fluorochrome conjugated αCD8, αCD44, and αCD62L and analyzed by flow cytometry.

Taking all of these data together, we believe that the three naive CD8⁺ T cell subsets analyzed in our study (CD5^{lo}, CD5^{hi}Ly6C⁻ and CD5^{hi}Ly6C⁺) are truly naive cells, which are different and clearly separated from MP cells both phenotypically and functionally, and that these naive subsets have distinct properties to differentiate into Tc17 cells.

2. The proliferation assay in vivo has demonstrated the difference among the three groups (extended data fig2a). The % of so call Fast cells, which means the proliferated cells upon antigen stimulation, showed a significant difference. Even for the "Slow" part, the MFI of each peak was shift, which means the proliferation upon the homeostatic stimulation was also not the same. So, the conclusion of "... their cell division profiles were also comparable" was not solid. Actually, the activation and proliferation issues could be quickly checked in vitro. It will be helpful to check the activation markers and CTV labelled cells in vitro, and checked in series time points.

We appreciate the reviewer for the valuable comments. As pointed out by the reviewer, the cell division in *Rag1*^{-/-} mice (Supplementary Fig. 2a in the revised manuscript) was observed in two forms, "fast" and "slow", and there was a difference in the frequency of each form of cell division. The "slow" form of division is known as "lymphopenia-induced homeostatic proliferation (HP)", which is induced by two major signals derived from contacts with both self-pMHC and IL-7, and we have previously reported that for naive CD8⁺ T cells, CD5^{hi} cells (particularly Ly6C⁺ subset) were superior to CD5^{lo} cells in this "slow" form of cell division (Cho et al, Immunity 2010). Therefore, we would like to emphasize that the frequencies shown in Supplementary Fig. 2a were based on the relative measurements, so that the relatively higher frequency of "slow" HP of CD5^{hi}Ly6C⁺ subset resulted inversely in a lower frequency of "fast" HP compared to that of CD5^{lo} subset (49.5% vs. 21.3% for "slow" HP and inversely 50.5% vs. 78.7% for "fast" HP).

The "fast" HP, unlike "slow" HP, seen in *Rag1*^{-/-} mice is driven by antigenic stimulation (derived from commensal bacteria), leading to rapid and robust cell division (indicated by full CTV dilution) as early as 3 days post-transfer. In this study, we observed no significant differences in the percentage and absolute cell numbers between three donor naive subsets of at least day 7 post-transfer (Fig. 2b in the revised manuscript), even though these subsets showed different abilities to produce IL-17 at this early time point (Supplementary Fig. 2a). Therefore, we believe that the observed differences in the IL-17-producing abilities between three naive subsets were not due to differences in their antigen-induced activation and subsequent proliferation particularly at an earlier time point of day 7 post-transfer. In the revised manuscript, we have now corrected the ambiguous description of the data in

Supplementary Fig. 2a to avoid any possible confusion and misleading (page 7, line 108–114 in the revised manuscript).

In addition, as suggested by the reviewer, we have also performed additional *in vitro* experiments to address whether there is any difference in the T cell activation and proliferation responses. For this, FACS-purified CTV-labeled CD5^{lo}, CD5^{hi}Ly6C⁻ and CD5^{hi}Ly6C⁺ naive CD8⁺ T cell subsets were stimulated with plate-bound anti-CD3/CD28 mAbs and analyzed at various time points (6, 24, 48 and 72 hours) for the activation and proliferation. As shown in **Reviewer Fig. 8**, overall expression patterns of various activation markers (CD25, CD44, CD62L and CD69) were similarly observed among three naive subsets from 6 to 72 hours after anti-CD3/CD28 stimulation, although CD25 expression at 6 hours post-stimulation appeared to be slightly delayed in CD5^{hi}Ly6C⁺ subset (due to its relatively lower sensitivity to TCR ligation compared to CD5^{lo} subset, as reported previously;

Cho et al, Nat Comm 2016).

Reviewer Fig. 8. Comparison of various activation marker expressions among three CD8⁺ T cell subsets. FACS-sorted three naive CD8⁺ T cell subsets were stimulated with plate-bound anti-CD3/CD28, and analyzed for the expression of various activation markers (CD25, CD69, CD44, and

CD62L) at different time points by flow cytometry.

In addition to the activation marker expressions, three naive subsets showed similar proliferative responses at 48 and 72 hours post-stimulation, as assessed by CTV dilution (Reviewer Fig. 9). Although overall cell division profiles were identical, we would like to point out that, when looking at the percentage of cells participating in the highest number of cell divisions at each time point (~2 and >5 divisions at 48 and 72 hours, respectively), a moderately increased proliferation was observed in CD5^{hi} subsets relative to CD5^{lo} subset. We assume that these increases might be due to the relatively higher sensitivity of CD5^{hi} subsets to IL-2 endogenously produced upon TCR stimulation compared to that of CD5^{lo} subset (Cho et al, Nat Comm 2016).

Reviewer Fig. 9. Comparison of proliferative responses among three CD8⁺ T cell subsets upon TCR stimulation. FACS-sorted, CTV-labeled naive CD8⁺ T cell subsets were stimulated with plate-bound anti-CD3/CD28. Cell proliferation was analyzed for CTV dilution by flow cytometry at 24, 48, and 72 hours after TCR stimulation.

Collectively, while we cannot entirely rule out any potential impact of subtle quantitative and/or qualitative differences of each naive subset under the *in vivo* conditions (i.e., *Rag1*^{-/-} recipients) that might not have been entirely recapitulated under these *in vitro* stimulation contexts, we believe that the three naive subsets were able to respond efficiently and sufficiently to antigenic stimulation *in vivo* to undergo robust antigen-driven proliferation, at least for the early stages of the responses. In addressing the valuable comments raised by the reviewer, we have now incorporated the aforementioned newly performed *in vitro* data (Reviewer Fig. 8 and 9; now added as Supplementary Fig. 2b,c in the revised manuscript with a statement on page 7, line 114–119).

3. Please note the problems with abbreviation: I) Some abbreviations were defined repeatedly in the text, such as ChIP, MFI, etc. II) The definition of some abbreviations is not written the full name, such as PBMC, MFI, etc. Please check and revise.

We apologize for the inconsistent use of some abbreviations. After a through review, we

have now corrected all of these in the revised manuscript.

4. Whether the relationship of IRF4 obtained in mice also applies to humans?

To address the reviewer's question, we FACS-isolated CD5^{lo} and CD5^{hi} subsets of human naive CD8⁺ T cells (CCR7⁺CD45RA⁺) from blood samples of healthy individuals. We then investigated the induction of IRF4 expression following 4 hours of plate-bound anti-CD3/CD28 stimulation. The results showed that, in human naive CD8⁺ T cells, IRF4 expression was higher in CD5^{lo} cells compared to CD5^{hi} cells, which was in agreement with the results obtained from murine naive CD8⁺ T cells (**Reviewer Fig. 10a, b**).

a

b

Reviewer Fig. 10. IRF4 expression of human naive CD8⁺ T cell subsets upon TCR stimulation. FACS-sorted human naive (CCR7⁺CD45RA⁺) CD8⁺ T cell subsets (CD5^{lo} and CD5^{hi}) from healthy individuals were stimulated with plate-bound anti-CD3/CD28 for 4 hours. **(a)** Gating strategy for FACS sorting (left) and sorting purity (right). **(b)** Expression of IRF4 shown in histogram (left) and MFI (right).

While these results were consistent with those from B6 CD8⁺ T cell subsets (higher IRF4 in CD5^{lo} subset than in CD5^{hi} subsets upon TCR stimulation; shown in Fig. 6a in the revised manuscript), it is important to note that our primary focus in this study lies on the endogenously expressed basal levels of SMAD3. For the sake of clarity, we have chosen not to incorporate these additional data in the revised manuscript, but instead presented here only for the reviewer's perusal.

5. Figure:

1) The scale of both HE and IF images are not clear. Fig. 1b, 2d, 3b, 3f, and 4d.

We appreciate the reviewer for these comments and have now added scale bars in all H&E and IF images in the revised manuscript.

2) The results of statistical analysis were not labeled in Fig. 4c, 5f, 7f, 7g, 7h, and 7i.

In the revised manuscript, we have now added statistical analysis to the indicated figures as advised by the reviewer.

3) Fig. 4b, why is body weight expressed in percentages instead of mass units (gram)? Or is it the change of body weight?

The reason we expressed body weight as a percentage in the indicated data was primarily to facilitate relative comparisons and changes. It is generally accepted that the use of percentages makes it easy to compare individual mice of different sizes and provide a standardized unit of measurement that is commonly used when expressing differences or changes in body weight. We also considered that using percentages would normalize the data, making it more intuitive and easier to interpret and communicate. And, most importantly, the weight change data using absolute mass units (grams) were the same as the percentage results, so we only presented the percentage data in the text. The absolute mass data are provided for the reviewer's perusal in **Reviewer Fig. 11**.

Reviewer Fig. 11. Body weight changes. Daily body weight changes in recipients of each naive CD8⁺ T cell subset shown in Fig. 4b in the revised manuscript was represented as absolute mass units (grams).

4) Fig. 6a, provide the column graph of MFI analysis of Fig. 6a left.

In the revised manuscript, we have added the column graph of MFI analysis to the

indicated figures as advised by the reviewer.

5) Fig. 6e and 6f, what did the "GFP-" and "GFP+" represent? It's confusing. If the vector has GFP fluorescent tag, the vector control should also be GFP+. Why choose GFP+ and GFP- for comparison?

As the reviewer pointed out, the retroviral vectors used in this study all contained GFP expression cassette. Overexpression or knockdown of *SMAD3* gene after retroviral transduction was assessed by detecting GFP expression using flow cytometry. Fig. 6e and 6f showed the results of transducing retroviral vectors containing either *SMAD3* gene or *SMAD3* shRNA. Based on the GFP expression after retroviral transduction, GFP⁺ cells (as cells transduced with retroviral vectors) were compared to GFP⁻ cells (as control cells untransduced within the same culture plate), resulting in a significant difference in Tc17 differentiation between GFP⁺ and GFP⁻ cells.

To confirm if these differences were not due to non-specific effects that might occur during viral transduction procedures, regardless of *SMAD3* overexpression or knockdown, we also conducted separately a control transduction experiment of using empty retroviral vectors, confirming no differences in Tc17 differentiation between GFP⁺ and GFP⁻ cells (Supplementary Fig. 5g,i). Therefore, we believe that comparing GFP⁺ cells directly with GFP⁻ cells under the same uniform culture condition minimize a potential variation in transfection efficiency and cell viability during viral transduction and subsequent *in vitro* activation and differentiation procedures.

6) Please provide the column graph of MFI analysis of Fig. 7a.

We have inserted the MFI analysis column graph in Figure 7a of the revised manuscript in response to the reviewer's comment.

7) Please add the R square of Fig. 7f and 7g.

We have added the R square in Figure 7f and 7g of the revised manuscript in response to the reviewer's comment.

6. Please provide information on the samples of the gene expression profile datasets GSE162335, GSE163314 and GSE151177.

We apologize for the limited information regarding the indicated gene expression profile datasets. In response to the reviewer's comment, we have now included detailed information on GSE162335, GSE163314, and GSE151177, and GSE116222 (newly added during this revision; see below our reply to Reviewer #2) in the Materials and Methods section of the revised manuscript as follows:

GSE162335: Transcriptome profiles of CD45⁺ cells from the lamina propria of 5 pouches without inflammation, 10 pouches with pouchitis, and 11 colons with ulcerative colitis.

GSE163314: Transcriptome profiles of CD45⁺ cells from colon biopsy tissue of 2 Crohn's disease patients, 2 Spondyloarthritis patients, and 2 Crohn's disease/Spondyloarthritis co-morbid patients.

GSE151177: Transcriptome profiles of inflammatory cells emigrating from a punch biopsy of 13 human Psoriasis lesional skin and the normal control skin of 5 healthy volunteers.

GSE116222: Transcriptome profiles of cells isolated from colonic biopsies collected from 3 healthy individuals and 3 patients with ulcerative colitis from an inflamed area of colon and adjacent non-inflamed area.

7. MATERIALS AND METHODS:

1) What is the method of IBD model establishment?

As an animal model for inducing IBD, we opted to adoptively transfer T cells into *Rag1*^{-/-} mice. This method has been previously utilized in several studies, where it was observed that the transfer of naive CD8⁺ T cells induced IBD, and this induction was dependent on the differentiation of IL-17-producing Tc17 cells (Tajima et al, J. Exp. Med. 2008). Therefore, in this study, we employed a model that allows us to observe potential differences in the ability of three distinct naive CD8⁺ T cell subsets to differentiate into Tc17 cells, with the aim of reflecting these differences in the manifestation of IBD.

This methodology was mentioned in the Materials and Methods section of the original manuscript. However, in response to the reviewer's comment, we have now added a more detailed description for IBD model in the revised manuscript.

2) What are the conditions for cell culture in vitro?

In response to the reviewer's question regarding the *in vitro* culture conditions, we would like to point out that information about all experiments conducted under these conditions were described in the Materials and Methods section of the original manuscript, under the following subheadings: "*In vitro* Tc17 cell polarization", "*In vitro* Th17/Tc17 cell polarization of human PBMC", "RNA sequencing and bioinformatics", "ChIP assay", and "Retroviral vectors for over-expression and short hairpin (shRNA) knock-down".

Nevertheless, addressing the reviewer's comments, we have now incorporated additional details about the experimental methods carried out in the *in vitro* culture conditions and have presented them more explicitly in the revised manuscript.

3) Methods and materials for ELISA are missing.

We apologize for omitting information about the experimental method for ELISA. This

information has now been included in the Materials and Methods section of the revised manuscript.

4) Please provide the probe sequence of qRT-PCR in this study.

We have incorporated the sequence information for the qRT-PCR probe in the Materials and Methods section of the revised manuscript.

5) Line 600-602, which parameter is used to determine negative and positive?

We utilized Seurat (R) for the analysis of single-cell RNA sequencing data generated by a droplet-based microfluidic system (10x Genomics). The droplet-based single-cell RNA sequencing method produces zero-inflated data, where most transcriptional expressions are zero, while actively expressed genes have values greater than zero. The expression data undergo preprocessing steps called "normalization" and "scaling". After these steps, we analyzed the expression of (A) *CD3E*, (B) *IL17A*, and (C) *SMAD3* (presented for the reviewer's perusal in **Reviewer Fig. 12**).

Reviewer Fig. 12. Analysis of *CD3E*, *IL17A*, and *SMAD3* using scRNA-seq data. Seurat R package-based analysis of single-cell RNA sequencing data generated by a droplet-based microfluidic system (10x Genomics). The droplet-based single cell RNA sequencing method generates zero-inflated data, meaning that most transcriptional expressions are 0, while genes that are certainly expressing will have over-zero values. The expression data undergo pre-processing steps namely "normalization" and "scaling". After these steps, we analyzed the expression of (a) *CD3E*, (b) *IL17A*, and (c) *SMAD3*. As shown in the figures, most cells have 0 expression, while some cells have meaningful expressions. The value 0.1 was selected to discrete "negative" and "positive" populations which is equal to "zero expression" and "over-zero expression", respectively.

As illustrated in Reviewer Fig. 12, the majority of cells exhibited zero expression, while some cells showed meaningful expressions. The threshold of 0.1 was chosen to distinguish between "negative" and "positive" populations, corresponding to "zero expression" and "over-zero expression", respectively. In response to the reviewer's comment, we have now incorporated this information in the Materials and Methods section of the revised manuscript.

6) Line 622-623, please provide the Addgene number of the plasmid used.

The information on the Addgene number of the plasmid used in this study (MigR1 and

pCMMP-LMP1-IRES-eGFP) has now been included in the Materials and Methods section of the revised manuscript.

7. Others:

1) Line 115, "proportions of IL-17A+IFN-g- (and IL-17A+IFN-g+) cells". Parentheses are redundant.

The parentheses have now been removed in the revised manuscript.

2) Line 501, "tranferred. into Rag1-/- mice." should be "transferred into Rag1-/- mice."

We apologize for the typo, and it has been corrected in the revised manuscript.

3) Line 515, "with anti-CD16/32; Fc block". Semicolon is redundant. And it should be Fc receptors (FcR) block rather than Fc block.

In response to the reviewer's suggestion, we have now corrected them in the revised manuscript.

Reviewer #2 (Remarks to the Author):

In the manuscript titled “Developmental self-reactivity determines the pathogenic Tc17 differentiation potential of naïve CD8 T cells by adjusting endogenous SMAD3 expression”, Lee et al report their identification of a subset of naïve T cells skewed for Tc17 differentiation based on CD5 expression level. The authors report that the different developmental capacity of the three naïve CD8 T cell subsets to undergo Tc17 differentiation in both IBD and GVHD murine models. Specifically, the authors demonstrate that the CD5^{lo} subset produces higher IL-17A which in turn exacerbates IBD and GVHD pathologies. Further, the authors show, using the IBD model, that migration of CD5^{lo} cells occurs through the CCR6 chemokine axis. Analysis of the cell-intrinsic mechanism for naïve T cell Tc17 skewing revealed that TCR strength/IL-2 signaling and SMAD3 expression are coupled to the generation of Tc17 cells in vivo and in vitro. Finally, using human T cells, the authors validated this novel immunological concept. Overall, this body of work robustly outlines the differential fate-skewing of naïve CD8 T cell subsets and the impact this can have on IBD and GVHD pathology in mouse systems. The overall study is designed very well including multiple mouse models of disease, different TCR tg model, as well as validation in human T cells. The mechanistic insights provided in this study are important but could be further strengthened with the inclusion of some additional data and controls described below.

Major points:

This study provides valuable insight into the different Tc17 differentiation capacity of CD5^{lo} naïve T cells in a SMAD3 dependent manner, and potential to establish autoimmunity in murine disease models. However, the link to human disease settings needs to be strengthened. The authors report a strong inverse relationship between IL-17A and SMAD3 expression in UC patient’s scRNA data (Fig7 k, l), but the entire premise of the paper is that the autoimmune association with IL17 comes from CD5^{lo} naïve cells. **This reviewer would like to know if the relationship between CD5 expression and IL17 expression also exists in the scRNA seq data.** Also, among the UC patients, only 18% of the CD3 T cells expressed either IL17A or SMAD3. **Can the authors show a correlation between disease severity and IL-17A or SMAD3 or CD5 levels?**

We appreciate the important questions raised by the reviewer. To investigate the correlation between *CD5* and *IL17A* expression, we reexamined the public scRNA-seq data from UC patients (GSE162335) utilized in our study. Although all T cells showed high levels of CD5 protein expression when analyzed by flow cytometry (**Reviewer Fig. 13a**), we noticed that this was not the case for the levels of CD5 mRNA expression. In fact, we observed that only approximately 10–16% of T cells (specifically, 10.9% for CD8⁺ T cells and 16.6% for CD4⁺ T cells) annotated from the analyzed scRNA-seq dataset were just identified as *CD5-positive* cells (**Reviewer Fig. 13b**). Similar data were observed in publicly available scRNA-seq datasets from other diseases analyzed in our study, such as Crohn’s

disease (CD), Spondyloarthritis (SpA), and Psoriasis (PS) (**Reviewer Figure 13c, d**).

Reviewer Fig. 13. CD5 expression in scRNA-seq data. (a) CD5 expression analyzed by flow cytometry. PBMC from healthy individuals were stained with fluorochrome-conjugated antibodies then analyzed for the expression of CD5 in CD4⁺ and CD8⁺ T cells, and non-T cells. (b–d) Following scRNA-seq data (GSE162335: lamina propria CD45⁺ cells from ulcerative colitis (UC) patients; GSE163314: CD45⁺ cells from colon of Crohn’s disease (CD) and spondyloarthritis (SpA) patients; GSE151177: Cells from skin biopsy of psoriasis (PS) patients) were downloaded from Gene Expression Omnibus and analyzed using Seurat R package. Cells were clustered using an unsupervised clustering method, then annotated according to the expression of key signature markers of CD8 T cells and CD4 T cells. All the other clusters were considered non-T cell clusters. CD5 expression were compared among the clusters.

Given the above observations that the scRNA-seq analysis may be relatively inefficient and suboptimal for determining the exact level of CD5 transcript expression, we suggest that examining the correlation between CD5 and IL17A (or SMAD3) expression under these conditions would be difficult to ensure accuracy. As expected, our reevaluation from the dataset of UC patients and other diseases did not reveal a significant correlation between CD5 and IL17A (or SMAD3) expression within the annotated T cell clusters (presented for the reviewer’s perusal in **Reviewer Fig. 14 and 15**); noted that the comparative analysis was performed for CD5-negative versus CD5-positive cells (rather than CD5^{lo} versus CD5^{hi} subset of CD5-expressing cells) due to insufficient levels of CD5 transcript expression. For clarity and simplicity, we have not added all these data, but instead have briefly mentioned this issue in the revised manuscript (page 20, line 407–409).

Reviewer Fig. 14. *IL17A* and *SMAD3* expression in *CD5+* and *CD5-* cells in UC and CD patients. (a–d) From scRNA-seq data of (a,b) UC patients and (c,d) CD patients, (a,c) *CD8* T cell clusters and (b,d) *CD4* T cell clusters were divided into *CD5+* (*CD5* expression > 0.1) and *CD5-* (*CD5* expression < 0.1) and compared for *IL17A* and *SMAD3* expression.

Reviewer Fig. 15. *IL17A* and *SMAD3* expression in *CD5+* and *CD5-* cells in SpA and PS patients. (a–d) From scRNA-seq data of (a,b) SpA patients and (c,d) PS patients, (a,c) *CD8* T cell clusters and

(b,d) *CD4* T cell clusters were divided into *CD5*⁺ (*CD5* expression > 0.1) and *CD5*⁻ (*CD5* expression < 0.1) and compared for *IL17A* and *SMAD3* expression.

With regard to the reviewer's second valid question regarding the correlation between disease severity and levels of *IL17A*, *SMAD3*, or *CD5*, we would like to clarify that the public scRNA-seq dataset of UC patients used in our study (GSE162335) did not include healthy controls and did not stratify stage-dependent disease severity and was therefore not appropriate to analyze the correlation between disease severity and relevant gene expressions questioned by the reviewer.

To answer the reviewer's question, we conducted an intensive search for public scRNA-seq datasets from other UC patient cohorts, but did not find suitable data analyzed by disease severity stage. Instead, we identified a public scRNA-seq dataset that analyzed healthy controls and UC patients (GSE116222; including inflamed and surrounding non-inflamed colon tissues; **Reviewer Fig. 16a**), and analyzed the following issues that we highlighted in our study: 1) the extent of *IL17A* and *SMAD3* gene expression between these groups; 2) the inverse relationship between *IL17A* and *SMAD3* gene expression; and 3) the association of *IL17A* or *SMAD3* gene expression in UC patients versus healthy individuals, perhaps most relevant to the reviewer's question.

Reviewer Fig. 16. Correlation between disease and *IL17A* in UC patients. **(a)** scRNA-seq data (GSE116222; Cells from a colon biopsy of a healthy donor, and two colon biopsies (uninflamed and inflamed area) of a UC patient. 3 replicates (Rep#1, Rep#2, Rep#3) of the experiment existed) were downloaded and analyzed. **(b,c)** T cell clusters were analyzed for *IL17A* and *SMAD3* expression. **(b)** Pie chart and **(c)** bar graph shows proportion of '*IL17A*⁺ or *SMAD3*⁺' (*IL17A*⁺*SMAD3*⁻, *IL17A*⁻*SMAD3*⁺, *IL17A*⁺*SMAD3*⁺) and '*IL17A*⁻ and *SMAD3*⁻' (*IL17A*⁻*SMAD3*⁻) cells. **(d)** '*IL17A*⁺ or *SMAD3*⁺' cells were divided into *IL17A*⁺ (*IL17A* expression > 0.1) and *IL17A*⁻ (*IL17A* expression < 0.1), then analyzed for

IL17A and *SMAD3* expression. (e,f) Ratio between *IL17A*⁺*SMAD3*⁻, *IL17A*⁻*SMAD3*⁺, and *IL17A*⁺*SMAD3*⁺ cells among '*IL17A*⁺ or *SMAD3*⁺' cells.

From the public scRNA-seq datasets analyzed, we found that the frequency of *IL17A* or *SMAD3* gene expression was approximately 5–12% of all *CD3*⁺ T cells annotated, and this low frequency was observed in both healthy individuals and UC patients (Reviewer Fig. 16b,c), which was similar to the results from UC patients described in our original manuscript (~17% of *CD3*⁺ T cells; Supplementary Fig. 6e). Furthermore, among these *IL17A*- or *SMAD3*-expressing T cells, we found a clear inverse relationship between *IL17A* and *SMAD3* gene expression (Reviewer Fig. 16d). Importantly, *IL17A* gene expression was only observed in UC patients and not in healthy individuals, and even within UC patients, it was higher in inflamed tissue biopsies compared to non-inflamed tissue biopsies (Reviewer Fig. 16e,f).

Similar to the increased frequency of *IL17A*-expressing T cells observed in UC patients compared to healthy individuals, the same trend was also observed in PS, CD, and SpA patients (Reviewer Fig. 17a–d).

Reviewer Fig. 17. Correlation between disease and *IL17A* in PS, CD, and SpA patients. (a–d) scRNA-seq data of (a,b) PS patients and (c,d) CD/SpA patients were analyzed. (a,c) Proportion of '*IL17A*⁺ or *SMAD3*⁺' (*IL17A*⁺*SMAD3*⁻, *IL17A*⁻*SMAD3*⁺, *IL17A*⁺*SMAD3*⁺) and '*IL17A*⁻ and *SMAD3*⁻' (*IL17A*⁻*SMAD3*⁻) cells. (b,d) Proportion of *IL17A*⁺*SMAD3*⁻, *IL17A*⁻*SMAD3*⁺, and *IL17A*⁺*SMAD3*⁺ cells

among *IL17A*⁺ or *SMAD3*⁺ cells.

Taken together, although we were unable to confirm that *IL17A* gene expression analyzed based on scRNA-seq directly correlated with stage-specific disease severity, we believe that the results obtained from our additional analysis provide some indication of an inverse correlation between *IL17A* and *SMAD3* gene expression in T cells and a potential link between *IL17A* gene expression in T cells and disease development based on the public scRNA-seq datasets of patients with various inflammatory diseases. For the sake of clarity, we prefer not to mention all these additional data, but instead have only added data from other UC patient cohorts (GSE116222) with relevant figure (**Reviewer Fig. 16d**; now added to **Supplementary Fig. 6g, top** in the revised manuscript) that further support our existing UC patient data (GSE162335).

Minor points:

1. In Figure 1, in mLN, three different naïve CD8 T cell subsets show a similar frequency and number with similar CTV levels. However, in the colon, they exhibited a different number of cells. Is this difference due to different proliferative or anti-apoptotic capacities among the cells?

The reviewer's comment seems to pertain to Fig. 2 and not Fig. 1 (Fig. 2b for day 7 mLN and Fig. 2d,e for day 7 colon). We appreciate the reviewer for these insightful questions, part of which was also asked by the reviewer #1 (possible difference in the proliferative capacity of each naïve subset). As the reviewer pointed out, the absolute cell numbers of the three naïve subsets in the mLN were identical at day 7 post-transfer; however, in the colon, CD5^{lo} subset was observed in higher numbers compared to CD5^{hi} subsets (Fig. 2b,e). During this revision, we confirmed that the reduced absolute cell number of CD5^{hi} subsets (particularly Ly6C⁺ subset) relative to CD5^{lo} subset in the colon but not mLN was not due to enhanced cell death of CD5^{hi} subsets compared to CD5^{lo} subset. In fact, we found that, when analyzed at day 14 after adoptive transfer into *Rag1*^{-/-} mice, both CD5^{lo} and Ly6C⁺ donor cells showed similar levels of Bim/Bcl2 and apoptotic cells (annexinV⁺PI⁺) (presented for the reviewer's perusal in **Reviewer Fig. 18**).

Reviewer Fig. 18. Comparison of apoptotic cell death between CD5^{lo} and Ly6C⁺ subsets. FACS-

sorted CD5^{lo} and Ly6C⁺ donor cells were transferred into *Rag1*^{-/-} mice, and isolated at day 14 from the large intestine (LI). Cells were cultured for 3 hours, and analyzed for anti- and pro-apoptotic Bcl2 and Bim expression, respectively, by flow cytometry. Apoptotic cell death was analyzed by annexinV and propidium iodide staining.

To further address a possible role of different proliferative capacities, we performed additional *in vivo* experiments where FACS-sorted three naive subsets were adoptively transferred into *Rag1*^{-/-} mice, pulsed with BrdU by giving them BrdU-containing drinking water for 2 days before analysis, and then analyzed at days 7 and 14 for BrdU uptake of each donor subset in the LI by flow cytometry (presented for the reviewer's perusal in **Reviewer Fig. 19, top**). The results revealed that at day 7 — a time point at which there were no signs of colitis onset — the proportion of BrdU⁺ cells among the three naive donor subsets was comparable, indicating equivalent cell proliferation capabilities during this early time period (**Reviewer Fig. 19, bottom left**). However, unlike a similar BrdU uptake at day 7, we found that proportion of BrdU⁺ donor cells at day 14 — a time point at which there were severe signs of colitis onset — was significantly higher in CD5^{lo} subset than CD5^{hi} subsets, indicating enhanced proliferative activity of CD5^{lo} subset (**Reviewer Fig. 19, bottom right**).

Reviewer Fig. 19. BrdU uptake of adoptively transferred naive CD8⁺ T cell subsets. BrdU incorporation analysis was performed as indicated in the experimental scheme (top), and cells isolated from the large intestine (LI) were stained with surface markers, then fixed and treated with DNase I. Cells were stained with fluorochrome-conjugated anti-BrdU antibody and analyzed by flow cytometry.

The aforementioned enhanced proliferative responses of the CD5^{lo} subset at day 14 appear to

be attributed to the more robust antigen exposure, presumably derived from the gut microbiota, under the severe inflammatory state and leakage in the colon. This is presumably because the CD5^{lo} subset triggers IL-17-induced inflammation and subsequent gut epithelial damage more rapidly than the CD5^{hi} subsets. Indeed, when we examined gut permeability by orally administering FITC-dextran to *Rag1*^{-/-} mice adoptively transferred with three naive subsets (presented for the reviewer's perusal in **Reviewer Fig. 20, top**), we found that mice transferred with CD5^{lo} subset had significantly higher gut damage compared to mice transferred with CD5^{hi} subsets, as evidenced by elevated levels of FITC-dextran in their serum at day 14, but not day 7 (**Reviewer Fig. 20, bottom**).

Reviewer Fig. 20. Comparison of gut permeability of adoptively transferred naive CD8⁺ T cell subsets. Gut permeability assay was conducted as indicated in the experimental scheme (top). The serum FITC concentration was measured using a spectrofluorometer.

These data therefore strongly support the notion that CD5^{lo} cells (which play the role of Tc17 cells) are able to reach the colon more quickly and efficiently, and gradually increase in sufficient numbers (perhaps between 7 and 14 days post-transfer) to initiate a faster IL-17-driven inflammation and subsequent gut epithelial damage.

Collectively, based on these additional *in vivo* data (and *in vitro* data described above in the reply to the reviewer #1), we believe that the difference in absolute cell numbers between mLN and colon observed at day 7 was not attributed to distinct activation/proliferation or varied survival rates. Instead, it was ascribed to differing gut-homing abilities, accompanied by distinct Tc17 differentiation potentials among the three naive subsets. We have now included these new data (**Reviewer Fig. 19 and 20; now added as Supplementary Fig. 2k,l**)

and corresponding paragraph in the Results section of the revised manuscript (page 9, line 147–159).

2. Clarity on whether the data came from the lymph node or colon would help the reader.

We appreciate the reviewer for bringing this to our attention, and have now clearly labeled the tissue analyzed in all relevant figures of the revised manuscript.

3. In Figure 3, authors show different migratory properties of the 3 naïve CD8 T cell subsets through a CCR6 dependent manner. How about CCR5 and GPR15? Is there any effect when authors use CCR5 or GPR15 KO CD8 T cell?

We appreciate the reviewer for raising this pertinent question. Investigating the link between gut-homing receptor expression and the induction of IBD was our top priority, as we observed no differences in the early (day 7 post-transfer) activation/proliferation of the three naïve subsets, but differences in early Tc17 differentiation and subsequent colonic migration capacity (Fig. 2). In this context, CCR6 expression differed significantly among the three naïve subsets (Fig. 3a), which correlated with their Tc17 differentiation potentials, and the role of CCR6 in rapid colonic migration and its association with enhanced IBD, particularly in the CD5^{lo} subset, was convincingly demonstrated using CCR6-deficient mice available in our animal colony.

However, we would like to emphasize that the above results do not rule out a role for other chemokine receptors, particularly CCR5 and GPR15, which we were unable to investigate due to the lack of mice lacking these receptors. In fact, studies in similar IBD models have highlighted the significance of various colonic homing receptors shown in mice lacking these receptors or being inhibited by pharmacological inhibitors (Mencarelli et al, Sci Rep 2016; Wang et al, Mucosal Immunol 2009; Habtezion et al, Gastroenterology 2016; Nguyen et al, Nat Immunol 2015; Allodi et al, Eur J Pharmacol 2023; Sacramento et al, BioRxiv 2023; and Seo et al, Cell Rep 2021). Specifically, reports indicating the inhibition of IBD by suppressing CCR5 expression or its function in T cells underscore the importance of these receptors. In this regard, marked delay in the early onset of IBD (i.e., day 21 but not day 14) seen in CCR6-deficient naïve donor subsets, particularly CD5^{lo} subset (Fig. 3f,g), suggests a potential involvement of other colonic homing receptors, such as CCR5, in this phenomenon.

To investigate this further, we performed an *in vitro* transwell migration experiments and examined whether the observed differences in CCR5 expression particularly between CD5^{lo} and Ly6C⁺ naïve donor subsets (collected from *Rag1*^{-/-} hosts at day 7 post-transfer) could impact their migration capabilities (presented for the reviewer's perusal in **Reviewer Fig. 21a**). As a result, we found that CD5^{lo} donor subset had significantly greater ability than Ly6C⁺ donor subset to trans-migrate in response to CCR6 ligand CCL20 and importantly CCR5 ligand CCL4 (**Reviewer Fig. 21b**), suggesting a role of both CCR6 and CCR5.

Reviewer Fig. 21. Comparison of chemotactic transwell migration ability of adoptively transferred naive CD8⁺ T cell subsets. (a) *In vitro* chemotactic transwell migration activity of CD5^{lo} and Ly6C⁺ subsets was performed as indicated in the experimental scheme. In brief, donor naive subsets were purified from mLN of adoptively transferred Rag1^{-/-} mice at day 7 post-transfer, and labeled with or without CTV. CD5^{lo} and Ly6C⁺ subsets were mixed at 1:1 ratio, and added to the upper compartment of transwell plates, while the lower compartment was added with CCL4 or CCL20. Cells in the upper and lower compartments were analyzed by flow cytometry. **(b)** Percentages of migrated cells; frequency of migrated cells was calculated as follow: bottom chamber counts/(bottom chamber counts + upper chamber counts) x 100.

Building on this *in vitro* observations, we also conducted additional *in vivo* adoptive transfer experiments (**Reviewer Fig. 22a**). In this experiment, Rag1^{-/-} mice were adoptively transferred with the two most distinct subsets, CD5^{lo} and Ly6C⁺ subset, and then injected with well-known CCR5 antagonist, Maraviroc, or with PBS as a control group. On day 21, we measured colonic migration capability of donor T cells and the severity of IBD symptom. The results showed that CCR5 inhibition significantly impaired colonic migration of Ly6C⁺ subset, although there was negligible difference in colonic migration of CD5^{lo} subset (**Reviewer Fig. 22b**). Similar to the impaired colonic migration data, the severity of IBD symptom was significantly reduced by *in vivo* CCR5 blockade particularly in mice receiving Ly6C⁺ subset, albeit a moderate decrease in mice receiving CD5^{lo} subset (**Reviewer Fig. 22c**). Similar findings were also observed for the absolute numbers of IL-17-producing cells (**Reviewer Fig. 22d**).

Reviewer Fig. 22. Role of CCR5 in the gut-homing ability and induction of IBD among naive CD8⁺ T cell subsets. (a) *In vivo* CCR5 blocking experiment was performed as indicated in the experimental scheme. (b) The migration efficiency between Maraviroc-injected and control groups was assessed in Rag1^{-/-} mice adoptively transferred with CD5^{lo} (left) and Ly6C⁺ (right) subsets. (c) Histological analysis data: representative H&E images (top), length of colon (bottom left), and histological score (bottom right). (d) Number of IL-17A⁺ donor cells in the large intestine.

These results imply some role of different levels of CCR5 expression in colonic migration and IBD induction of different naive subsets, particularly of Ly6C⁺ subset. Nevertheless, however, the observed effect of CCR5 blockade was somewhat less pronounced in CD5^{lo} subset, suggesting a complementary role of other colonic homing receptors, such as CCR6, which was expressed higher in CD5^{lo} subset than CD5^{hi} subsets. Collectively, as pointed out by the reviewer, we believe that, in addition to CCR6, differences in CCR5 expression may also contribute to differences in the ability of three naive subsets to migrate to the colon and induce IBD. For the sake of clarity, we did not add the results of all the experiments we performed to answer the reviewers' questions, but presented them here only for the reviewer's perusal. Instead, in the revised manuscript, we added *in vitro* transwell migration data (Reviewer Fig. 21; now added as Supplementary Fig. 3b,c) indirectly supporting a role of additional chemokine receptors such as CCR5, along with a statement suggesting the

potential involvement of other gut-homing receptors besides CCR6 (**page 9, line 165–167; page 10, line 171–177; and page 11, line 193–196**).

4. In Figure 3g, h, it would be helpful to add the CCR6+ control group.

We appreciate the reviewer for this suggestion. IBD caused by adoptively transferring naive CD8⁺ T cell subsets from normal wild-type (WT) mice (*CCR6*^{+/+}) into *Rag1*^{-/-} mice already shows severe diarrhea and colonic inflammation by day 14. Therefore, we would like to point out that due to animal ethics regulations, most experiments in our study had an end point of day 14–15 post-transfer, except for some cases where IBD symptoms occurred at more delayed time points. However, unlike WT donor cells, CCR6 knockout (KO) donor cells did not show any such symptoms on day 14, but rather maintained a near-normal colonic tissue integrity (Fig. 3b,c). Therefore, we terminated the experiments on day 14 for normal WT donor cells, with the exception of CCR6-deficient donor cells, which we extended for another 7 days to examine for signs of IBD. As a result, we found that CCR6-deficient donor cells, particularly CD5^{lo} subset, showed IBD symptoms with diarrhea and colonic tissue inflammation at day 21 (Fig. 3f,g).

Although it would be useful to add the data of WT donor cells at day 21, we suggest that the role of CCR6 — namely promoting rapid and efficient colonic migration of donor cells, particularly CD5^{lo} subset — can be sufficiently explained by our prior data of day 14 showing the clear difference between the WT and KO cells (Fig. 3b,c).

REVIEWERS' COMMENTS

Reviewer #1 (Remarks to the Author):

The authors added sufficient experiments, showing data to answer all of my concerns.

Reviewer #2 (Remarks to the Author):

The authors were very responsive to the prior comments and have thoroughly addressed all outstanding questions. This reviewer has no further questions.

Reply to the Reviewers' comments:

Reviewer #1 (Remarks to the Author):

The authors added sufficient experiments, showing data to answer all of my concerns.

Reviewer #2 (Remarks to the Author):

The authors were very responsive to the prior comments and have thoroughly addressed all outstanding questions. This reviewer has no further questions.

We appreciate the reviewers' valuable comments and questions and the reviewers agreed that all issues being raised have been thoroughly addressed in the revised manuscript.